# Process-based modelling to evaluate simulated groundwater levels and frequencies in a Chalk catchment in Southwest England

Simon Brenner[1], Gemma Coxon[2,4], Nicholas J. K. Howden[3,4], J. Freer[2,4] and Andreas Hartmann[1,3]

[1]Institute of Earth and Environmental Sciences, Freiburg University, Germany
[2]School of Geographical Sciences, University of Bristol, Bristol, UK
[3]Department of Civil Engineering, University of Bristol, Bristol, UK
[4]Cabot Institute, University of Bristol, Bristol, UK

*Correspondence to*: S. Brenner (simon.brenner@hydrology.uni-freiburg.de)

## Abstract

Chalk aquifers are an important source of drinking water in the UK. Due to their properties, they are particularly vulnerable to groundwater related hazards like floods and droughts. Understanding and predicting groundwater levels is therefore important for effective and safe water management. Chalk is known for its high porosity and, due to its dissolvability, exposed to karstification and strong subsurface heterogeneity. To cope with the karstic heterogeneity and limited data availability, specialised modelling approaches are required that balance model complexity and data availability. In this study, we present a novel approach to evaluate simulated groundwater level frequencies derived from a semi-distributed karst model that represents subsurface heterogeneity by distribution functions. Simulated groundwater storages are transferred into groundwater levels using evidence from different observations wells. Using a percentile approach we can assess the number of days exceeding or falling below selected groundwater level percentiles. Firstly, we evaluate the performance of the model to simulate groundwater level time series by a spilt sample test and parameter identifiability analysis. Secondly, we apply a split sample test on the simulated groundwater level percentiles to explore the performance in predicting groundwater level exceedances. We show that the model provides robust simulations of discharge and groundwater levels at three observation wells at a test site in a chalk-dominated catchment in Southwest England. The second split sample test also indicates that percentile approach is able to reliably predict groundwater level exceedances across all considered time scales up to their 75th percentile. However, when looking at the 90th percentile, it only provides acceptable predictions for the long time periods and it fails when the 95th percentile of groundwater exceedance levels is considered. Modifying the historic forcings of our model according to expected future climate changes, we create simple climate scenarios and we show that the projected climate changes may lead to generally lower groundwater levels and a reduction of exceedances of high groundwater level percentiles.

## 1 Introduction

The English Chalk aquifer region extends over large parts of south-east England and is an important water resource aquifer, providing about 55 % of all groundwater-abstracted drinking water in the UK (Lloyd, 1993). As a carbonate rock the English Chalk is exposed to karstification, i.e. the chemical weathering (Ford and Williams, 2007), resulting in particular surface and subsurface features such as dolines, river sinks, caves and conduits (Goldscheider and Drew, 2007; Gutiérrez et al., 2014; Stevanovic, 2015). Consequently, karstification also produces strong hydrological subsurface heterogeneity (Bakalowicz, 2005). The interplay between diffuse and concentrated infiltration and recharge processes, as well as fast flow through karstic conduits and diffuse matrix flow, result in complex flow and storage dynamics (Hartmann et al., 2014a). Even though Chalk tends to less intense karstification, for instance compared to limestone, its karstic behaviour has increasingly been recognised (Maurice et al., 2006, 2012; Fitzpatrick, 2011).

Apart from the good water quality, favourable infiltration and storage dynamics which make chalk aquifers a preferred source of drinking water in the UK, their karstic behaviour also increases the risk of fast drainage of their storages by karstic conduit flow during dry years. This also increases the risk of groundwater flooding as a result of fast responses of groundwater levels to intense rainfalls due to fast infiltration and groundwater recharge processes. Groundwater flooding, i.e. when groundwater levels emerge at the ground surface due to intense rainfall (Macdonald et al., 2008), tend to be more severe in areas of permeable outcrop like the English Chalk (Macdonald et al., 2012) as also experienced repeatedly in other karst areas in Europe (Parise, 2003, 2010; Bonacci et al., 2006; Jourde et al., 2007; Gutiérrez, 2010; Naughton et al., 2012; Parise et al., 2015). Groundwater drought indices tend to be more related to recharge conditions in Cretaceous Chalk aquifers than in granular aquifers (Bloomfield and Marchant, 2013). Due to the fast transfer of water from the soil surface to the main groundwater system, chalk aquifers tend to be more sensitive to external changes, as for instance shown by Jackson et al. (2015) who found significant groundwater level declines in 4 out of 7 chalk boreholes in a UK-wide study using historic groundwater level observations.

Climate projections suggest that the UK will experience increasing temperatures, with less rainfall during the summer but warmer and wetter winters (Jenkins et al., 2008). This may stress these groundwater resources, and increase the risk of groundwater droughts and potentially winter groundwater flooding. For those reasons, assessment of potential future changes in groundwater dynamics, concerning groundwater droughts, median groundwater levels as well as groundwater flooding is broadly recommended and is subject of current research around the world (Naughton et al., 2012, 2015; Jackson et al., 2015; von Freyberg et al., 2015; Jimenez-Martinez et al., 2016; Moutahir et al., 2017; Perrone and Jasechko, 2017).

However, present approaches mostly rely on statistical distribution functions to express groundwater dynamics and groundwater level exceedance probabilities (e.g., Bloomfield et al., 2015; Kumar et al., 2016) and it is questionable whether the shapes of these distribution functions remain the same when climate or land use change. Physics-based hydrological simulation models that incorporate hydrological processes in a relatively high detail can be considered to potentially provide the most reliable predictions, especially under a changing environment. However, there are considerable limitations in obtaining the necessary information to estimate the structure and the model parameters, especially for subsurface processes, and this inevitably increases modelling uncertainties (Perrin et al., 2003; Beven, 2006).

The definition of appropriate model structures and parameters from limited information becomes problematic when modelling karst aquifers. In order to achieve acceptable simulation performance they have to include representations of karstic heterogeneity in their structures. Distributed karst modelling approaches are able to simulate groundwater levels on a spatial grid but their data requirements mostly limit them to theoretical studies (e.g., Birk et al., 2006; Reimann et al., 2011) or well explored study sites (e.g., Hill et al., 2010; Jackson et al., 2011; Oehlmann et al., 2014). Lumped karst modelling approaches consider physical processes at the scale of the entire karst system. Although they are strongly simplified, they can include karst peculiarities such as different conduit and matrix systems (Maloszewski et al., 2002; Geyer et al., 2008; Fleury et al., 2009). Since they are easy to implement and do not require spatial information, they are widely used in karst modelling (Jukić and Denić-Jukić, 2009). Simple rainfall-runoff models with more than 5-6 parameters are often regarded to end up in equifinality (Wheater et al., 1986; Jakeman and Hornberger, 1993; Ye et al., 1997), i.e. their parameters lose their identifiability (Wagener et al., 2002; Beven, 2006). For that reason, recent research took advantage of auxiliary data, such as water quality data or tracer experiments (Hartmann et al., 2013b; Oehlmann et al., 2015a). These studies showed that adding such information allows identifying the necessary model parameters, therefore enabling the model to reflect the relevant processes.

Up to now, most lumped karst models have been applied for rainfall-runoff simulations. Groundwater levels were simulated in quite a few studies (Adams et al., 2010; Ladouche et al., 2014; Jimenez-Martinez et al., 2016), however mostly relying on

very simple representation of karst hydrological processes and disregarding the scale discrepancy between borehole (point scale) and modelling domain (catchment scale) at which they were applied.

In this study, we present a novel approach to predict and evaluate groundwater level frequencies in chalk-dominated catchments. It uses a previously developed semi-distributed process-based model (VarKarst, Hartmann et al., 2013b) that we further developed to simulate groundwater levels. To assess groundwater level frequencies we formulated a percentile of groundwater based approach that quantifies the probability of exceeding or falling below selected groundwater levels. We exemplify and evaluate our new approach on a Chalk catchment in Southwest England that had to cope with several flooding events in the past. Finally we apply the approach on simple climate scenarios that we create by modifying our historic model forcings to show how changes in evapotranspiration and precipitation can affect groundwater level frequencies.

## 2 Study site and data availability

Located in West Dorset in the south-west of England the river Frome drains a rural catchment with an area approximately 414 km² (Figure 1). The catchment elevation varies from over 200 m above sea level (a.s.l.) in the north-west to sea level in the south-east. The topography is very flat with a mean slope of 3.9 % and a mean height of approximately 111 m a.s.l.. The climate can be defined as oceanic with mild winters and warm summers (Dorset County Council, 2009). Howden (2006) characterised the Frome as highly groundwater-dominated. During the summer months, discharge of the Frome typically is very low, hardly reaching 5 m³/s (Brunner et al., 2010). The geology is predominated by the Cretaceous Chalk outcrop which underlays around 65 % of the catchment. The headwaters of the Frome include outcrops of the Upper Greensand, often overlain by the rather impermeable Zig-Zag Chalk (Howden, 2006). The middle reaches of the Frome traverse the Cretaceous Chalk outcrop followed by Palaeogene strata in the lower reaches, eventually draining into Poole Harbour. The major aquifer Chalk appears mainly unconfined. However, in the lower reaches it is overlain by Palaeogene strata, resulting in confined aquifer conditions. The region around the Frome catchment is known for the highest density of solution features in the UK (Edmonds, 1983) which can be mainly observed in the interfluve between the Frome and Piddle (Adams et al., 2003). Loams over chalk, shallow silts, deep loamy, sandy and shallow clays constitute the primary types of soils occurring in the study area (Brunner et al., 2010). The soils of the upper parts of the catchment are mainly shallow and well drained (NRA, 1995). In the middle and lower reaches the soils are becoming more sandy and acidic due to waterlogged conditions caused by either groundwater or winter flooding (NRA, 1995; Brunner et al., 2010). Due to its geological setting, the area is prone to groundwater flooding. It has occurred several times at different locations, for example in Maiden Newton during winter 2000/2001 (Environment Agency, 2012) and in Winterbourne Abbas during summer 2012 (Bennett, 2013).

**Figure 1: Location map with an overview on the Frome catchment**

## 3 Methodology

In order to consider karstic process behaviour in our simulations we use the process-based karst model VarKarst introduced by Hartmann et al. (2013b). VarKarst includes the karstic heterogeneity and the complex behaviour of karst processes using distribution functions that represent the variability of soil, epikarst and groundwater and was applied successfully at different karst regions over Europe (Hartmann et al., 2013a, 2014b, 2016). We use a simple linear relationship that takes into account effective porosities and base level of the groundwater wells (see Eq. 1) enabling the model to simulate groundwater levels based on the groundwater storage in VarKarst. Finally, a newly developed evaluation approach is used by transferring simulated groundwater level time series into groundwater level frequency distributions and comparing them to observed behaviour at a number of monitored wells.

### 3.1 The model

The VarKarst model operates on a daily time step. Similar to other karst models, it distinguishes between three subroutines representing the soil system, the epikarst system and the groundwater system but it also includes their spatial variability , which is expressed by distribution functions that are applied to a set of $N$=15 model compartments (Figure 2). Pareto functions as distribution functions have shown to perform best in previous work (Hartmann et al., 2013a, 2013b), as well as the number of 15 model compartments (Hartmann et al., 2012). Including the spatial variability of subsurface properties in this manner, the VarKarst model can be seen as a hybrid or semi-distributed model. All relevant model parameters are provided in Table 1. For a detailed description of VarKarst see the appendix or Hartmann et al. (2013b).

**Figure 2: The VarKarst model structure**

The model was driven by two input time series (Precipitation and Potential Evapotranspiration (PET)), and the 13 variable model parameters (see Table 2) were calibrated and evaluated by four observed time series (discharge and the three boreholes, see subsection 3.3). Similar to Kuczera and Mroczkowski (1998) we use a simple linear homogeneous relationship which translates the groundwater storage [mm] into a groundwater level [m a.s.l.]:

$$h_{GW}(t) = \frac{V_{GW,i}(t)}{1000 * p_{GW}} + \Delta h \tag{1}$$

The related parameters are $h_{gw}$ [m] and $p_{gw}$ [-]. $h_{gw}$ is the difference of the base of the contributing groundwater storage (that is simulated by the model) and the base of the well that is used for calibration and evaluation. $p_{gw}$ represents the average porosity of the rock that is intersected by the well.

### 3.2 Data availability

The daily discharge data for gauge East Stoke was obtained from the Centre for Ecology & Hydrology (CEH, http://nrfa.ceh.ac.uk/ ) and dates back to the 1960s. The borehole data was provided by the Environment Agency (EA) and obtained via the University of Bristol. The total data used for modelling in this study can be seen in Table 1. The three boreholes (Ashton Farm, Ridgeway and Black House) comprised high resolution raw data which had been collected at a 15-minute interval. For further analysis, the data were aggregated to daily time averages. The potential evapotranspiration has a strong annual cycle. Since most recent data from years 2009-2012 was missing, representative PET-years were calculated on the basis of the last fifty years. Climate projections were obtained from the UK Climate Projections User Interface (UKCP09 UI, http://ukclimateprojections-ui.metoffice.gov.uk/ ). For more information about the UKCP see Murphy et al. (2010).

### 3.3 Model calibration and evaluation

We use the Shuffled Complex Evolution Method (SCEM) for our calibration, which is based on the Metropolis-Hastings algorithm (Metropolis et al., 1953; Hastings, 1970) and the Shuffled Complex Evolution algorithm (SCE, Duan et al., 1992). The Metropolis-Hastings algorithm uses a formal likelihood measure and calculates the ratio of the posterior probability densities of a "candidate" parameter set that is drawn from a proposal distribution and a given parameter set. If this ratio is larger or equal than a number randomly drawn from a uniform distribution between 0 and 1, the "candidate" parameter set is accepted. This procedure is repeated for a large number of iterations. If the proposal distribution is properly chosen, the Markov Chain will rapidly explore the parameter space and it will converge to the target distribution of interest (Vrugt et al.,

2003). In the SCEM algorithm, "candidate" parameter sets are drawn from a self-adapting proposal distribution for each of a predefined number of clusters. Again a random number [0,1] is used to accept or discard "candidate" parameter sets. The SCEM algorithm was applied in default mode using the Gelman-Rubin convergence criteria (Vrugt et al., 2003). In our study, we use the Kling-Gupta efficiency (KGE; Gupta et al., 2009) as objective function, which can be regarded as an informal likelihood measure, or more generally a monotonically increasing performance metric of model skill (Smith et al., 2008). It was chosen by trial and error comparing the simulation performances during calibration and validation obtained with different objective functions (RMSE and other). We found that we obtain the most robust results with the KGE. To decide whether to accept or discard a parameters set, we compare the KGEs of the "candidate" and the given parameter sets. Such procedure was already applied in various studies (Engeland et al., 2005; Blasone et al., 2008; McMillan and Clark, 2009) and is possible if the error functions are monotonically increasing with improved performance. We achieved this in the SCEM algorithm by defining KGE as:

$$\text{KGE} = -\sqrt{(r-1)^2 + (\alpha-1)^2 + (\beta-1)^2} \qquad (2)$$

$$\alpha = \frac{\sigma_s}{\sigma_o} ; \beta = \frac{\mu_s}{\mu_o}$$

With r as the linear correlation coefficient between simulations and observations, and $\sigma_s$ $\sigma_o$ and $\mu_s$ $\mu_o$ as the means and standard deviations of simulations and observations, respectively.

The posterior parameter distributions derived from SCEM provide information about the identifiability of the parameters. The more they differ from a uniform posterior distribution the higher the identifiability of a model parameter. We present different calibration distributions to show the use of auxiliary data for parameter identifiability.

Parameter ranges were chosen following previous experience with the VarKarst model (Hartmann et al., 2013a, 2013b, 2014b, 2016). Besides the quantitative measure of efficiency, a split sample test (Klemeš, 1986) was carried out. Our data covered precipitation, evapotranspiration, discharge and groundwater levels from 2000 to the end of 2012. We calibrated the model on the period 2008-2012 and used the period 2003-2007 for validation. We chose this reversed order to be able including the information of 3 boreholes that was only available for 2008-2012. Three years were used as warm-up for calibration and validation, respectively. During calibration, the most appropriate of the $N$=15 groundwater compartments to represent each groundwater well was found by choosing the compartment with the best correlation to the groundwater dynamics of the well.

This procedure was repeated for each well and each Monte Carlo run and finally provides the three model compartment numbers that produce the best simulations of groundwater levels at the three operation wells and the best catchment discharge according to our selected weighting scheme. During calibration, we used a weighting scheme which was found by trial and error, as we stepwise added borehole data to our discharge observations. Discharge and the borehole at Ashton Farm were both weighted as one third as Ashton farm is located in the lower parts within the catchment while the other two boreholes were located at higher elevation at the catchment's edge and weighted one sixth each. In order to explore to contribution of the different observed discharge and groundwater time series during the calibration, we use SCEM to derive the posterior parameter distributions using (1) the final weighting scheme, (2) only discharge, (3) only Ashton farm, and (4) only the other two boreholes (equally weighted). Posterior parameter distributions are plotted as cumulative distributions. The more parameters that show sensitivity, the more information is contained in the selected calibration scheme.

## 3.4 The percentile approach

Even though the VarKarst model includes spatial variability of system properties by its distribution functions, its semi-distributed structure does not allow for an explicit consideration of the locations of ground water wells. Its model structure allowed for an acceptable and stable simulation of groundwater level time series of the three wells (see subsection 4.1), but for groundwater management, frequency distributions of groundwater levels, calculated over the time scale of interest, are commonly preferred. For that reason we introduced a groundwater level percentile based approach. Other than Westerberg et al. (2016) that transferred discharge time series into signatures derived from flow duration curves, we calibrate directly with the discharge and groundwater time series in order to evaluate the performance of our approach for selected time periods (see evaluation below). Similar to the calculation of standardised precipitation or groundwater indices (e.g., Lloyd-Hughes and Saunders, 2002; Bloomfield and Marchant, 2013), we create cumulative frequency distributions of observed groundwater levels and the simulated groundwater levels from the previously evaluated model. Now, the exceedance probability or percentile for a selected observed groundwater level (for instance, the groundwater level above which groundwater flooding can be expected) can be used to define the corresponding simulated groundwater level and the number of days exceeding or falling below the chosen groundwater level can directly be extracted from the frequency distributions (Figure 3). Note that this procedure is performed after the model is calibrated and validated with KGE as described in the previous subsection. We avoided a calibration directly to the flow percentiles, as temporal information would have been removed, which would have resulted in a lower prediction performance of the model.

**Figure 3: Schematic description of the percentile approach**

As the approach is meant to be applied in combination with climate change scenarios, we perform an evaluation on multiple time scales and flow percentiles. We assess the 5[th], 10[th], 25[th], 50[th], 75[th], 90[th] and 95[th] percentiles on temporal resolutions of years, seasons, months, weeks and days. The deviation between modelled and observed number of exceedance days of these different percentiles is quantified by the **m**ean **a**bsolute **d**eviation (MAD) between simulated exceedances (SE) and observed exceedances (OE):

$$MAD_p = mean\left(abs\left(\sum SE_{i,x} - \sum OE_{i,x}\right)\right) \qquad [d] \tag{3}$$

Where $x$ stands for the time scale (years, months, weeks, days) and $p$ is the respective percentile. To better compare the deviation for different percentiles we normalize the MAD to a **p**ercentage of mean **a**bsolute **d**eviation (PAD) with the total number of days of the chosen time scale:

$$PAD_p = \frac{MAD_p}{dp_x} * 100 \qquad [\%] \tag{4}$$

where $dp_x$ is a normalizing constant standing for the total number of days of the respective time scale and percentile. For example, if we take the time scale *months* and the *75[th] percentile* of exceedances we got a $dp_x$ of (100-75) % x (365.25 / 12) days. To evaluate the prediction performance of the approach, percentiles are derived from the daily data of the calibration period and then applied on the validation period similar to the split sample test in subsection 3.3. In this way we are able to evaluate our model over different thresholds and in terms of temporal resolution.

**3.5 Establishment of simple climate scenarios and assessment of groundwater level frequency distributions**

Given the model performance assessment above, we then use our approach to assess future changes of groundwater level
frequencies at our study site. We derive projections of future precipitation and potential evapotranspiration by manipulating
our observed 'baseline' climate data. We extract distributional samples of percentage changes of precipitation and
evaporation from the UK probabilistic projections of climate change over land (UKCP09) for (1) a low emission scenario
and (2) a high emission scenario for the time period of 2070-2099. This enables us to capture, in a pragmatic and
computationally efficient approach, for the two emission scenarios the general range of changes for the most pertinent
variables that we think will most impact changes to monthly-seasonal GW responses. We focus on projected median delta
values for change in mean temperature (°C) and precipitation (%) as well as the respective $25^{th}$ and $75^{th}$ percentile from the
probabilistic projections and apply them on our input data. For our model input we transfer projected temperatures into
evapotranspiration via the Thornthwaite equation (Thornthwaite, 1948). In this way, we obtain 3 x 3 projections (3x
precipitation and 3x evapotranspiration) for each of the emission scenarios that also address the uncertainty associated with
the projections. The resulting simulations will provide an estimate of possible future changes of groundwater level
frequencies for the two emission scenarios including an assessment of their uncertainty.

**4   Results**

**4.1  Model calibration and evaluation**

Table 2 shows the optimised parameter values as well as the model performance. The simulation of the discharge shows
KGE values of 0.73 and 0.58 in the calibration and validation period, respectively. The borehole simulations show high KGE
values and only slight deteriorations in the validation period. The parameters are located well within their pre-defined
ranges. Mean soil storage $V_{mean,S}$ and mean epikarst storage $V_{mean,E}$ are 2015.6 mm and 1011.7 mm, respectively. The
porosity parameter at Ashton Farm is the highest, followed by the borehole at Black House. Ridgeway shows the smallest
porosity value. For Ashton Farm and Blackhouse the calibration chose the groundwater storage compartment 7, whilst for
Ridgeway it chose the compartment number 8.
Figure 4 plots the observations against simulations for the calibration and validation period. Modelled discharge generally
matches the seasonal behaviour of the observations. However, some low-flow peaks are not depicted well in the simulation.
When looking at the groundwater levels, the simulation of Ashton Farm appears to be most adequate. However, there are
considerable periods when differences from the observations can be found for all wells. Simulations at Ridgeway and Black
House show moderate performance in capturing peak groundwater levels. Notably the simulation at Black House is slightly
better in the validation period. The cumulative parameter distributions derived by SCEM indicate that the model parameters
were well identifiable when we use all available data (Figure 5), while some parameters remain hardly identifiable when
only parts of the available data were used for calibration. Here identifiability of parameters is simply the extent that the
cumulative parameter distributions span the sampled parameter limits, where highly constrained or near optimal is classed as
identifiable. For instance, when only discharge was used for calibration (green lines), the parameters related to groundwater
(porosity $p_{GW}$ and groundwater level offset $\Delta h$) happen to be unidentifiable. In addition, the groundwater parameters are only
identifiable when their respective time series is considered (i.e. the yellow and blue lines at $p_{GW,A}$ and $\Delta h_{GW,A}$). In turn, the
epikarst storage $V_{meanE}$ is not identifiable when only the groundwater well data is used (yellow and red lines). We also note,
as we would expect, that the final cumulative parameter distributions occur in different parts of the parameter space
depending on the combination of performance metrics from different observations.

**Figure 4: Modelled discharge [m³/s] of the Frome at East Stoke and groundwater levels [m a.s.l.] at the boreholes Ashton Farm, Ridgeway and Black House**

**Figure 5: Cumulative parameter distributions (blue) of all model parameters; strong deviation from the 1:1 (dark grey) indicate good identifiability**

### 4.2 The percentile approach

When simulated peak values of groundwater levels are compared to the observations, we find a rather moderate agreement. Using the percentile approach we find different thresholds to exceed our selected groundwater level percentiles. This is elaborated for the 90th percentile of simulated and observed groundwater levels of Ashton farm (Figure 6).

**Figure 6: Illustration of the percentile approach. Time series of the observed (grey dots) and modelled (green line) groundwater level at Ashton Farm. The dotted lines represent the respective 90th percentile**

Table 3 shows the mean observed and modelled exceedances of all selected thresholds (the 5th, 10th, 25th, 50th, 75th, 90th, and 95th percentiles) at all temporal resolutions in the validation period. By comparing matches in the number days of exceedance we evaluate our model at different percentiles and time scales. The left value is the mean absolute deviation (MAD) and the right value is the percentage of absolute deviation (PAD). We can see that the higher the percentile, the larger is the deviation between observed and modelled exceedances. The same is true for the PAD when moving from lower to higher temporal resolutions. The MAD gets lower with higher temporal resolution.

**Table 4: Deviations of simulated to observed exceedances of different percentiles in the validation period (borehole: Ashton Farm). The left value is the mean absolute deviation MAD [d], the right value is the deviation percentage PAD [%]**

### 4.3 Impact of simulated climate changes on groundwater level distributions

The results of applying the two climate projections to the model can be found at Table 4 and in Figure 7. They display the mean model outputs (Qsim, AET) and mean exceedances per year, calculated on the basis of our modelled time series. Both emission scenarios (low & high) lead to an increased modelled actual evapotranspiration and to decreased discharge simulations. In addition, both emission scenarios show a substantial reduction in exceedances of high percentiles. We also find that the standard error of the exceedances and non-exceedances of high emission scenario tends to be higher than the standard error of the low emission scenario.

**Figure 7: Mean model input (mm/a), mean modelled output (mm/a) and mean (non-)exceeded percentiles (number/a) in the reference period and both scenarios (borehole: Ashton Farm; future period: 2070-2099). The circles indicate the spread among the 9 realisations for each of the two scenarios**

**Table 4: Model output and (non-)exceedances of percentiles in the reference period and the two scenarios (borehole: Ashton Farm, time period 2070-2099)**

# 5 Discussion

## 5.1 Reliability of the simulations

A decrease of simulation performance in the validation period is normally to be expected because there is always a tendency to compensate for structural limitations and observational uncertainties during the calibration. The low decrease in model performance from 11% (groundwater prediction at Black House, $KGE_{GW,B}$) to 21% (discharge prediction, $KGE_Q$) during the validation period indicates a certain robustness of the calibrated parameters and is comparable to split sample tests in other studies (e.g., Parajka et al., 2007; Perrin et al., 2001) although we have to acknowledge that for other applications a higher degree of robustness may be required. In addition, it is corroborated by their generally mainly high identifiability derived by SCEM for the final calibration scheme that used all 4 available observed discharge and ground water level time series. Applying the Shuffled Complex Evolution Metropolis algorithm and step wise increasing the calibration data (only discharge, only groundwater, all together), we show that discharge data alone, as well as groundwater data alone, do not provide enough information to identify all of our model parameters as the posteriors of some of the model parameters remain close to a uniform distribution. This is similar to the work of Schoups and Vrugt (2010) who found unidentifiable parameter values with their models calibrating only against discharge. The different calibration schemes visualised in the cumulative parameter distributions show that initially unidentifiable parameters become identifiable when the related time series is considered. Using all information, all model parameters are identifiable, which is in accordance with preceding research that showed the usefulness of multi-objective approaches. For instance, Kuczera and Mroczkowski (1998) demonstrated that a combination of groundwater and discharge observations can reduce parameter uncertainty. As we were mostly focussing on the difference among the calibration steps with increasing data, the use of KGE as an informal likelihood measure seems justifiable.

A look at the parameter values reveals an adequate reflection of the reality. However, $V_{mean,S}$ and $V_{mean,E}$ are quite high considering that initial ranges for these parameters were 0-250/0-500 mm (Hartmann et al., 2013a, 2013c). As previous studies took place in fairly dry catchments, the ranges were extended substantially to deal with the wetter climate in southern England. A high $a_{SE}$ indicates a high variability of soil and epikarst thicknesses favouring lateral karstic flow concentration (Ford and Williams, 2007). Butler et al. (2012) notes that the unsaturated zone of the Chalk is highly variable, ranging from almost zero near the rivers to over 100 m in interfluves.

Additionally, the mean epikarst storage coefficient $K_{mean,E}$ is quite low, indicating fast water transport from the epikarst to the groundwater storage which is in accordance to other studies (e.g., Aquilina et al., 2006). The value of parameter $a_{fsep}$ indicates that a significant part of the recharge is diffuse. A moderately high conduit storage coefficient $K_C$ and a high $a_{GW}$ indicate that there is a significant contribution of slow pathways by the matrix system. A rather low value but sensitive of $K_C$ was found when calibrating only by discharge operations, indicating some interactions of $K_C$ with other model parameters (Saltelli et al., 2008). This is in accordance with the findings by Jones and Cooper (1998) as well as by Reeves (1979) who reported 30 % and 10-20 % of the recharge occurring through (macro-) fissures in Chalk catchments, respectively. Although groundwater flow in the chalk is dominated by the matrix, given antecedent wet conditions, fracture flow can increase significantly (Lee et al., 2006; Ireson and Butler, 2011; Butler et al., 2012b). Overall, split-sample test, parameter identifiability analysis, realistic values of parameters and plausible simulation results provide strong indication for a reliable model functioning.

## 5.2 Performance of the percentile approach

Based on the idea of the standardised precipitation or groundwater indices (Lloyd-Hughes and Saunders, 2002; Bloomfield and Marchant, 2013) our percentile approach permits to improve the performance of the model to reflect observed

groundwater level exceedances. It yields acceptable performance for years to days up to the 90[th] percentile. A reduction of precision with the time scale is obvious but in an acceptable order of magnitude when the validation period is considered. Although deviations are considerable both in the calibration and validation period, they are stable demonstrating certain robustness but also the limitations of our approach. Although the variable model structure of the VarKarst model was shown to provide more realistic results than commonly used lumped models (Hartmann et al., 2013b) it still simplifies a karst system's natural complexity. This can be seen in the simulated time series at Ashton Farm and Black House, which indicate an over-estimation of high levels and an under-estimation of low levels. The reason for this behaviour might be due to the modelling assumption of a constant vertical porosity, despite the knowledge that there can be a strongly non-linear relation between chalk transmissivity and depth. Several studies acknowledge that hydraulic conductivity in the Chalk follows a non-linear decreasing trend with depth (Allen et al., 1997; Wheater et al., 2007; Butler et al., 2009). This is mainly attributed to the decrease of fractures because of the increasing overburden and absence of water level fluctuations (Williams et al., 2006; Butler et al., 2012a). Hydraulic conductivities in the Chalk can span several orders of magnitude (Butler et al., 2009) and are particularly enhanced at the zone of water table fluctuations (Williams et al., 2006). In addition, cross-flows occurring in the aquifer can lead to complicated system responses in the Chalk (Butler et al., 2009). For the sake of a parsimonious model structure, these characteristics were omitted in this study but their future consideration could help to improve the simulations if information about the depth profile of permeability is available. Such decrease of performance was also found for standardised indices that use probability distributions instead of a simulation model (Vicente-Serrano et al., 2012; Núñez et al., 2014; Van Lanen et al., 2016). To improve the approach's reliability for higher groundwater level percentiles, a model calibration that is more focussed on the high groundwater level percentiles may be a promising direction. A consideration of the time spans above the 90[th] percentile will allow for a better simulation quality. This could be further evaluated by using different percentile weighting schemes, stepwise increasing the weight on the target percentile.

## 5.3 Applicability and transferability of our approach

We prepared two scenarios by manipulating our input data using probabilistic projections of annual changes of precipitation and potential evaporation at 2070-2099 for a low and a high emission scenario. This may neglect some of the changes on climate patterns predicted by climate projections but it is based on local and real meteorological values of the reference period, therefore avoiding problems that arise when historic and climate projection data show pronounced mismatches during their overlapping periods. Our results revealed that both scenarios lead to less exceedances over higher percentiles and more non-exceedances of lower percentiles indicating a higher risk of groundwater drought at our study site. However, one problem that arises from our approach is that we do not consider changes in the seasonal patterns of our input variable, for example the increase of winter precipitation. If this increase was considered, the results would probably yield more exceedances of higher percentiles, as for instance found by Jimenez-Martinez et al. (2015). The purpose of the simple climate scenarios was to provide an application example of the new methodology, which is rather hypothetical considering the large uncertainties of current climate projections. We believe that our 9 realisations are sufficient to show that different possible future changes have a non-linear impact on groundwater level frequencies. Although quite simplistic, our results are qualitatively in accordance with previous studies indicating increased occurrence of droughts in the UK (Burke et al., 2010; Prudhomme et al., 2014). The risk of drought occurrences might increase depending on the magnitude of change in evapotranspiration. However, more research and the application of more elaborated scenarios is necessary to completely understand the consequences of the change in groundwater frequency patterns in the UK chalk regions.

As the VarKarst model is a process-based model that includes the relevant characteristics of karst systems over range of climatic settings (Hartmann et al., 2013b) our approach can to some extent be used to assess future changes of groundwater level distributions and also be applied in other regions. This may bring some advantage concerning approaches that used

transfer functions (Jimenez-Martinez et al., 2016) or regression models (Adams et al., 2010) for estimating groundwater levels, if enough data for model calibration and evaluation is available.

As has been noted by Cobby et al. (2009), the likelihood and depth of groundwater inundations is one of the major challenges for future research of groundwater flooding. Since it is a lumped approach it may provide, after Butler et al. (2012), "a good indication of the likelihood of groundwater flooding, but do[es] not indicate where the flooding will take place". A spatial determination of the groundwater table as in Upton and Jackson (2011) would be possible but only in catchments where the borehole network is extensive. Thereby, the possibility to model several boreholes with one single calibration, due to compartment structure in VarKarst, might be also an advantage. Butler et al. (2012) noted that the parameterization of the unsaturated zone is a major difficulty in the Chalk. Since this study struggles also with the porosity, future work should take a closer look at this subject.

## 6    Conclusions

We used an existing process-based lumped karst model to simulate groundwater levels in a chalk catchment in Southwest England. Groundwater levels were simulated by translating the modelled groundwater storage into groundwater levels with a simple linear relationship. To evaluate our approach we analysed the agreement of observed and simulated groundwater level exceedances for different percentiles. Finally, a simple scenario analysis was undertaken to investigate the potential future changes of groundwater level frequencies that affect the risk of groundwater flooding, as well as the risk of groundwater droughts. The model performance for discharge and the groundwater levels was satisfying showing the general adequacy of the model to simulate groundwater levels in the chalk. It also revealed shortcomings concerning higher groundwater levels. This was corroborated by the percentile approach that showed a robust performance up to the 90[th] percentile. A scenario analysis using UKCP projections on expected regional climate changes showed that expected changes may lead to an increased occurrence of low groundwater levels due to increasing actual evaporation. Overall, our study shows that semi-distributed process-based modelling can be a valuable tool to simulate and predict groundwater frequencies in Chalk regions where information is too limited for the application of distributed models. Here, a thorough model evaluation is essential to obtain reliable and consistent results. In order to obtain more reliable results we recommend collecting more data about the hydrogeological properties of our study site to improve the structure of our model regarding the porosity and the unsaturated zone. In addition, longer time series and an adapted calibration approach which, in particular, emphasizes on the >90[th] percentiles of groundwater levels could significantly improve our simulations. In addition we propose to apply the method on other catchments to test the transferability of our approach and to quantify the variability of climate change impacts over a wide range of Chalk catchments across the UK.

## Acknowledgements

This publication contains Environment Agency information © Environment Agency and database right. Thanks to Dr Jens Lange and Dr Sophie Bachmair, University of Freiburg, for their valuable advice. Support for GC, JF and NH was provided by NERC MaRIUS: Managing the Risks, Impacts and Uncertainties of droughts and water Scarcity, grant number NE/L010399/1. The article processing charge was funded by the German Research Foundation (DFG) and the University of Freiburg in the funding programme Open Access Publishing.

## 7    Appendix

Within the VarKarst model, the parameter $V_{mean,S}$ [mm] and the distribution coefficient $a_{SE}$ [-] define the variation of soil storage capacities across the $N$ model compartments. They are used to calculate the soil storage capacity $V_{S,i}$ [mm] for every

compartment $i$ by Eqs. (3,4) in Table 5. We apply the same distribution coefficient $a_{SE}$ when we derive the epikarst storage distribution by the mean epikarst depth $V_{mean,E}$ [mm] (Eqs. (6,7) in Table 5). We determine actual evapotranspiration from each soil compartment $E_{act,i}$ is calculated by reducing potential evapotranspiration, which is found by the Thornthwaite equation (Thornthwaite, 1948), by the soil saturation deficit (Eq. (1) in Table 5). Surface runoff is found by the excess of soil and epikarst storage of the previous model compartment (Eq. (2) in Table 5). With surface runoff and actual evapotranspiration know, the stored water volume at each soil compartment $V_{Soil,i}$ [mm] can be calculated by simply applying water balance.

The recharge from the soil to the epikarst $R_{Epi,i}$ [mm] is calculated by the excess of the soil storage (Eq. (5) in Table 5), while the epikarst outflow follows a linear storage assumption (Eqs. (8,9) in Table 5). Again, water balance allows determining the stored water $V_{Epi,i}$ [mm] at each time step $t$ and each epikarst compartment $i$. The downward flux from the epikarst considers a diffuse ($R_{diff,i}$ [mm]) and concentrated groundwater recharge ($R_{conc,i}$ [mm]) component that are found by a variable separation factor $f_{C,i}$ [-] and a distribution coefficient $a_f$ [-] (Eqs. (10,11,12) in Table 5). The diffuse component recharges the groundwater compartments beneath the respective epikarst layers ($i = 1…N$-1). The concentrated component flows laterally to compartment $i = N$ and therefore recharges the conduit system.

Similar to the epikarst compartment, variable groundwater storage coefficients $K_{GW,i}$ [d] are calculated (Eq. (15) in Table 5) and applied to calculate the discharges of the matrix system (Eq. (13) in Table 5) and the conduit system (Eq. (14) in Table 5), which together sum up to the entire discharge of the system (Eq. (15) in Table 5). Knowing groundwater recharge and groundwater discharge for each model compartment $i$ again allows determining the stored volume of water within the groundwater compartment $V_{GW,i}$ at time step $t$, which is used to simulate the groundwater levels (Eq. (1) in subsection 3.1).

**Table 5: Model routines, variables and equations solved in the VarKarst model**

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

1    **10   Figures**

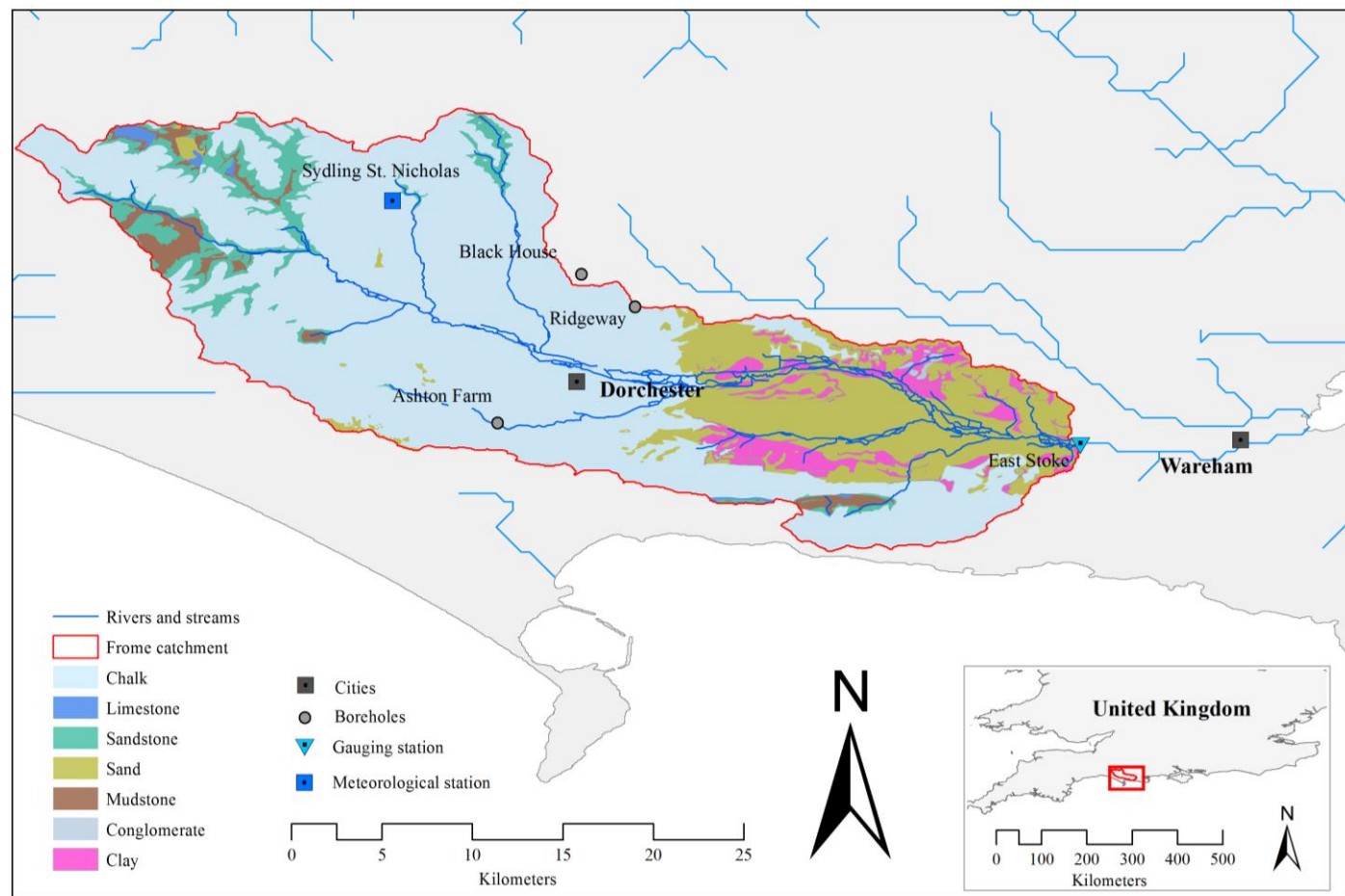

4    **Figure 1: Location map with an overview on the Frome catchment**

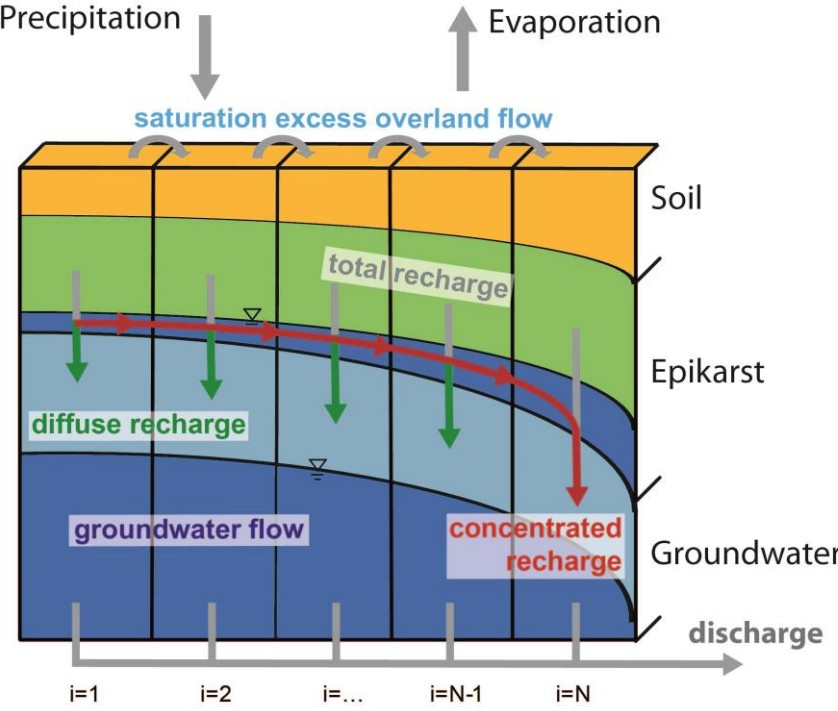

3  **Figure 2: The VarKarst model structure**

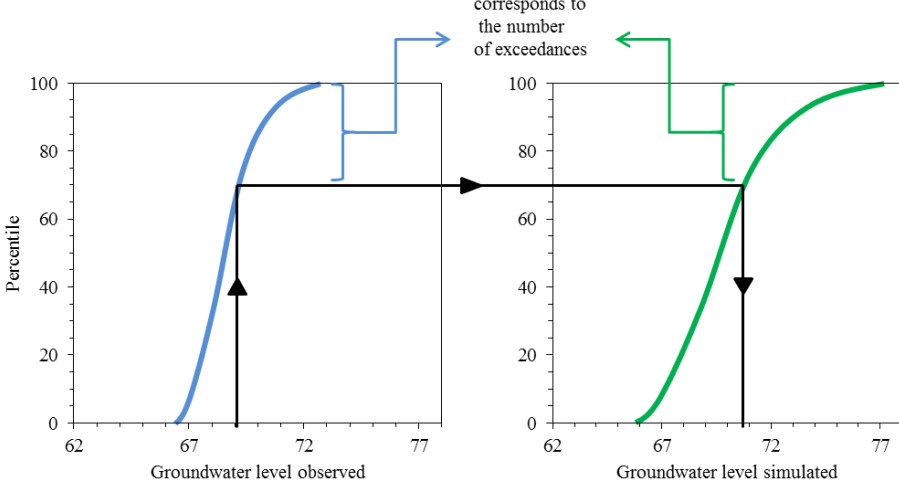

6  **Figure 3: Schematic description of the percentile approach**

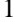

Figure 4: Modelled discharge [m³/s], and groundwater levels [m a.s.l.] at the boreholes Ashton Farm, Ridgeway and Black House

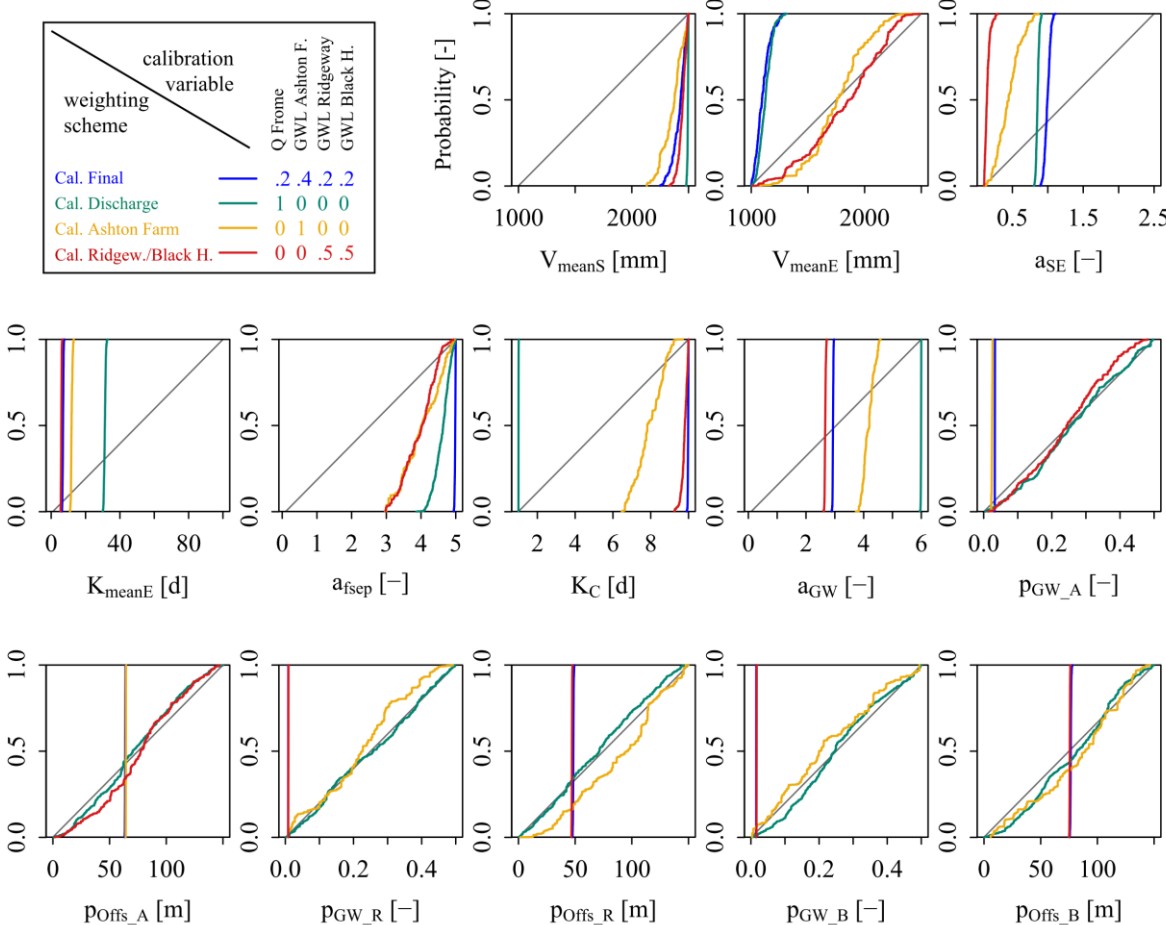

Figure 5: Cumulative parameter distributions (blue) of all model parameters; strong deviation from the 1:1 (dark grey) indicate good identifiability

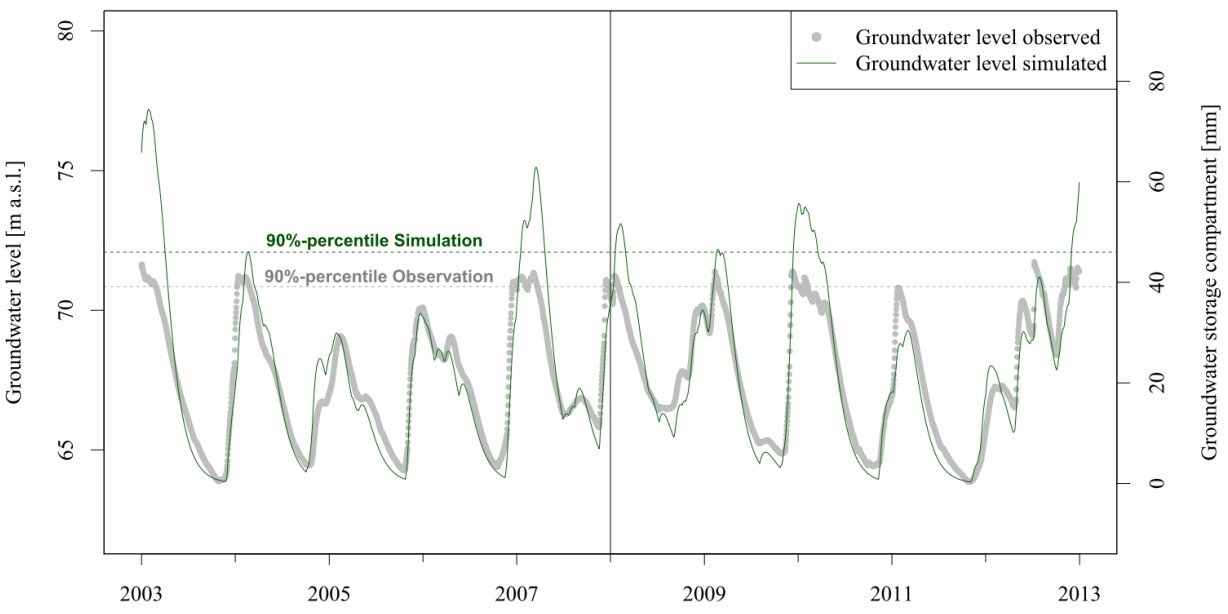

Figure 6: Illustration of the percentile approach. Time series of the observed (grey dots) and modelled (green line) groundwater level at Ashton Farm. The dotted lines represent the respective 90th percentile

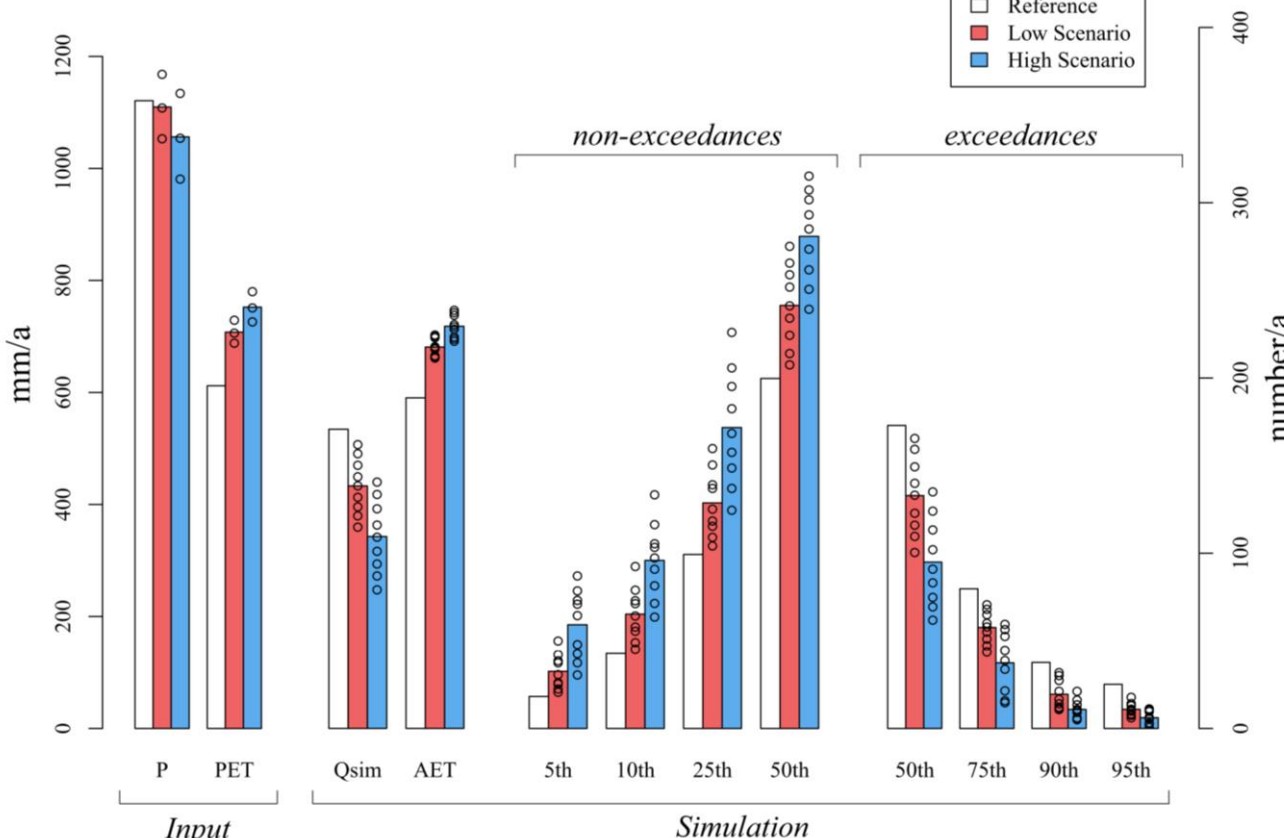

3 **Figure 7: Mean model input (mm/a), mean modelled output (mm/a) and mean (non-)exceeded percentiles (number/a) in the**
4 **reference period and both scenarios (borehole: Ashton Farm; future period: 2070-2099). The circles indicate the spread among the**
5 **9 realisations for each of the two scenarios**

1 **11  Tables**

3 **Table 1: All available data used in the study**

| Parameter | Station | Source | Period of time | Resolution | Unit |
|---|---|---|---|---|---|
| Precipitation | Sydling St. Nicolas (44006) | CEH | 01.01.2000-31.12.2012 | daily | mm d$^{-1}$ |
| Discharge | East Stoke (44001) | CEH | 01.01.2000-31.12.2012 | daily | m$^3$s$^{-1}$ |
| Pot. Evapotranspiration | Catchment Cut East Stoke | CEH | 01.01.2000-31.12.2008 | daily | mm d$^{-1}$ |
| Groundwater Levels | Ashton Farm, Ridgeway, Black House | EA | 01.01.2003-31.12.2012 | daily | m a.s.l. |
| Climate Delta values | Grid Box Nr. 1698 (25*25 km) | UKCP | 2070-2099 | annual | °C, % |

 **Table 2: Model parameters, descriptions, ranges and optimised values**

| Parameter | Description | Unit | Ranges | | Weighting | Optimised Values |
|---|---|---|---|---|---|---|
| | | | Lower | Upper | | |
| $V_{mean,S}$ | Mean soil storage capacity | mm | 1000 | 2500 | | 2015.6 |
| $V_{mean,E}$ | Mean epikarst storage capacity | mm | 1000 | 2500 | | 1011.7 |
| $K_{mean,E}$ | Epikarst mean storage coefficient | d | 0.1 | 2.5 | | 0.7246 |
| $K_C$ | Conduit storage coefficient | d | 1 | 100 | | 38.722 |
| $a_{fsep}$ | Recharge separation variability constant | - | 0.1 | 5 | | 1.1864 |
| $a_{GW}$ | Groundwater variability constant | - | 1 | 10 | | 5.9966 |
| $a_{SE}$ | Soil/epikarst depth variability constant | - | 0.1 | 6 | | 1.8928 |
| $p_{GW,A}$ | Ashton Farm groundwater level porosity parameter | - | 0.001 | 0.5 | | 0.0069 |
| $\Delta h_{GW,A}$ | Ashton Farm groundwater level offset parameter | m | 0 | 150 | | 64.167 |
| $p_{GW,R}$ | Ridgeway groundwater level porosity parameter | - | 0.001 | 0.5 | | 0.0016 |
| $\Delta h_{GW,R}$ | Ridgeway groundwater level offset parameter | m | 0 | 150 | | 48.718 |
| $p_{GW,B}$ | Black House groundwater level porosity parameter | - | 0.001 | 0.5 | | 0.0032 |
| $\Delta h_{GW,B}$ | Black House groundwater level offset parameter | m | 0 | 150 | | 78.448 |
| | | | | | | |
| $KGE_Q$ | Model performance for discharge | - | 0 | 1 | 0.2 | 0.73/0.58* |
| $KGE_{GW,A}$ | Model performance for groundwater level at Ashton Farm | - | 0 | 1 | 0.4 | 0.94/0.80* |
| $KGE_{GW,R}$ | Model performance for groundwater level at Ridgeway | - | 0 | 1 | 0.2 | 0.86/ - * |
| $KGE_{GW,B}$ | Model performance for groundwater level at Black House | - | 0 | 1 | 0.2 | 0.83/0.74* |

*Calibration/validation.

**Table 3: Deviations of simulated to observed exceedances of different percentiles in the validation period (borehole: Ashton Farm). The left value is the mean absolute deviation MAD [d], the right value is the deviation percentage PAD [%]**

| Time period | Percentiles | | | | | | |
|---|---|---|---|---|---|---|---|
| | 5 | 10 | 25 | 50 | 75 | 90 | 95 |
| **5 years** | 5.00 / 0.29 | 30.00 / 1.83 | 38.00 / 2.77 | 16.00 / 1.75 | 1.26 / 5.04 | 19.00 / 10.40 | 90.00 / 98.56 |
| **years** | 2.60 / 0.75 | 13.60 / 4.14 | 14.40 / 5.26 | 21.20 / 11.61 | 4.33 / 17.30 | 19.80 / 54.21 | 26.00 / 142.37 |
| **year-seasons** | 0.65 / 0.75 | 4.10 / 4.99 | 3.60 / 5.26 | 6.90 / 15.11 | 6.74 / 26.94 | 6.45 / 70.64 | 6.50 / 142.37 |
| **months** | 0.22 / 0.75 | 1.37 / 4.99 | 1.20 / 5.26 | 2.73 / 17.96 | 7.94 / 31.76 | 2.58 / 84.87 | 2.23 / 146.75 |
| **weeks** | 0.05 / 0.74 | 0.33 / 5.27 | 0.27 / 5.18 | 0.61 / 17.36 | 7.82 / 31.27 | 0.58 / 83.56 | 0.54 / 153.10 |
| **days** | 0.01 / 0.75 | 0.05 / 5.35 | 0.04 / 5.26 | 0.09 / 17.96 | 7.94 / 31.76 | 0.08 / 84.88 | 0.08 / 159.91 |

**Table 4: Model output and (non-)exceedances of percentiles in the reference period and the two scenarios (borehole: Ashton Farm, time period 2070-2099)**

| Scenario | Qsim | AET | 5th | 10th | 25th | 50th | 75th | 90th | 95th |
|---|---|---|---|---|---|---|---|---|---|
| | mm/a | mm/a | non exc/a | non exc/a | non exc/a | exc/a | exc/a | exc/a | exc/a |
| Reference | 534 | 590 | 17.6 | 41.3 | 95.6 | 172.9 | 79.7 | 37.7 | 25.2 |
| Low | 433 | 681 | 31.4 | 62.8 | 123.9 | 132.9 | 57.6 | 19.5 | 10.9 |
| High | 343 | 718 | 57.0 | 92.3 | 165.3 | 94.9 | 37.5 | 10.9 | 6.1 |

**Table 5: Parameters, descriptions and equations solved in the VarKarst model**

| Model routine | Variable | Description | Equation | Unit | Eq. Nr. |
|---|---|---|---|---|---|
| Soil | $E_{act,i}(t)$ | Actual evapotranspiration | $= E_{pot}(t) \dfrac{\min[V_{Soil,i}(t) + P(t) + Q_{Surface,i}(t),\ V_{S,i}]}{V_{S,i}}$ | mm d$^{-1}$ | (1) |
| | $Q_{Surf,i+1}(t)$ | Surface flow to the next model compartment | $= max[V_{Epi,i}(t) + R_{Epi,i}(t) - V_{S,i},\ 0]$ | mm d$^{-1}$ | (2) |
| | $V_{max,S}$ | Maximum soil storage capacity | $= V_{mean,S}\, 2^{\left(\frac{a_{SE}}{a_{SE}+1}\right)}$ | mm | (3) |
| | $V_{S,i}$ | Soil storage distribution | $= V_{max,S}\left(\dfrac{i}{N}\right)^{a_{SE}}$ | mm | (4) |
| | $R_{Epi,i}(t)$ | Recharge to the epikarst | $= max[V_{Soil,i}(t) + P(t) + Q_{Surface,i}(t) - E_{act,i}(t) - V_{S,i},]$ | mm d$^{-1}$ | (5) |
| Epikarst | $V_{max,E}$ | Maximum epikarst storage capacity | $= V_{mean,E}\, 2^{\left(\frac{a_{SE}}{a_{SE}+1}\right)}$ | mm | (6) |
| | $V_{E,i}$ | Epikarst storage distribution | $= V_{max,E}\left(\dfrac{i}{N}\right)^{a_{SE}}$ | mm | (7) |
| | $Q_{Epi,i}(t)$ | Outflow of the epikarst | $= \dfrac{\min[V_{Epi,i}(t) + R_{Epi,i}(t) + Q_{Surface,i}(t),\ V_{E,i}]}{K_{E,i}}\Delta t$ | mm d$^{-1}$ | (8) |
| | $K_{E,i}$ | Epikarst storage coefficient | $= K_{max,E}\left(\dfrac{N-i+1}{N}\right)^{a_{SE}}$ | d | (9) |
| | $R_{diff,i}(t)$ | Diffuse recharge | $= f_{C,i} Q_{Epi,i}(t)$ | mm d$^{-1}$ | (10) |
| | $R_{conc,i}(t)$ | Concentrated recharge | $= (1 - f_{C,i})\, Q_{Epi,i}(t)$ | mm d$^{-1}$ | (11) |
| | $f_{C,i}$ | Recharge separation factor | $= \left(\dfrac{i}{N}\right)^{a_{fsep}}$ | - | (12) |
| Groundwater | $Q_{GW,i}(t)$ | Groundwater contributions of the matrix | $= \dfrac{V_{GW,i}(t) + R_{diff,i}(t)}{K_{GW,i}}$ | mm d$^{-1}$ | (13) |
| | $Q_{GW,N}(t)$ | Groundwater contribution of the conduit system | $= \dfrac{\min[V_{GW,N}(t) + \sum_{i=1}^{N} R_{conc,i}(t),\ V_{crit,OF}]}{K_C}\Delta t$ | mm d$^{-1}$ | (14) |
| | $K_{GW,i}$ | Variable groundwater storage coefficient | $= K_C\left(\dfrac{N-i+1}{N}\right)^{-a_{GW}}$ | d | (15) |
| | $Q_{main}(t)$ | Discharge | $= \dfrac{A_{max}}{N}\sum_{i=1}^{N} Q_{GW,i}(t)$ | l s$^{-1}$ | (16) |