# Peer review of "Process-based modelling to evaluate simulated groundwater levels and frequencies in a Chalk catchment in Southwest England"

_Natural Hazards and Earth System Sciences, 2016_

## Short Comment (SC1)

*This manuscript describes the application of an existing hydrologic model for karst aquifers. The approach of evaluating the model calibration in terms of hydrologic exceedance rates appears to be a new and useful approach. Exceedance rates of projected hydrologic simulations are evaluated for human safety or the needs of species (e.g., https://pubs.er.usgs.gov/publication/sir20145089). Therefore, if a model is to be used for this purpose, it makes good sense to evaluate the model directly on the basis of exceedance frequencies.*

We thank Dr Andy Long for his positive evaluation and his valuable and helpful comments.

*Main comments:*

1. *The abstract explains that the approach to simulate groundwater level frequency is novel. I would say that this is not the novel part, because the time-series records were simulated and simply converted into frequency distributions, which is a common way to summarize hydrologic time-series records. However, the novel part is that the model calibration is evaluated on the basis of frequency distributions, which I have not seen before, and I suggest presenting it that way. The tile is more accurate: "A percentile approach to evaluate simulated groundwater levels and frequencies. . ."*

We will change the title and the abstract accordingly.

2. *Section 5.2 discusses the possibility of focusing the calibration on high percentiles. I don't totally agree that longer time-series records would be needed to do this, and this section could benefit from further discussion of this idea. For example, an approach could be developed to evaluate the usefulness and data adequacy of such an endeavor. You could vary the weights within the observed time-series record for individual observations at different exceedances to tailor the calibration to a target percentile. It would be possible to calibrate to a different weighting scheme for each percentile. Further, an uncertainty analysis could be applied on each separate calibration run, and quantifying the presumed decrease in uncertainty as the percentile increases could be useful. Then, when you make predictions for different percentiles, you can also report the differences in uncertainty. This idea also applies to section 5.3, which discusses the prediction of increased drought.*

We will take these considerations into account and incorporate a more detailed discussion on high percentile calibration at the end of Section 5.2.

3. *The Introduction discusses risks to events such as groundwater flooding and drought. I suggest adding a short statement to this effect in the Abstract to emphasize the need for this study in terms of natural hazards.*

We will add a statement in the abstract.

*Other comments:*

1. *p. 2, lines 24-26: indicates that karst groundwater levels were simulated by lumped models only in a few instances, but see also Long and Derickson (1999), Long and Mahler (2013), and Pinault et al. (2001).*

We thank the referee for this useful suggestions. We will add Long and Mahler (2013) and (

Derickson and Long (1999) to the literature comparison. We could not find any simulation of groundwater levels in the work of Pinault et al. (2001) and we added the work of Kumar et al. (2016) instead.

2. *p. 3, lines 22-23: describes a new approach to show groundwater levels as frequency distributions. Showing hydrologic time-series data as frequency distributions is a common method. Please explain how this is new, or describe it differently.*
3. *p. 3, line 35: "PET" should be defined.*

We agree with both comments and we will change the text accordingly.

4. *p. 4, line 34: discusses a "weighting scheme." I think the calibration weights are applied to observations, but that should be explained here for clarity.*

A clearer description of the model and parameters estimation will be provided in the revised manuscript (see also specific comment 1 of the other referee).

5. *figure 7: what is the meaning of "manipulated" in the caption?*

This refers to the manipulation of our observed "baseline" data, see chapter 3.5. We will clarify this in the caption of figure 7 in the revised manuscript.

6. *table 5: I think these result apply to a particular model time step (e.g., daily), but I'm not sure. Please clarify.*

These are the mean model outputs (Qsim, AET) and exceedances per year in the simulation period 2070-2099 which was calculated on the basis of our model which runs at a daily time step. We will clarify this in the revised version.

**References**

Derickson, R.., Long, a. ., 1999. Linear systems analysis in a karst aquifer. J. Hydrol. 219, 206–217. doi:10.1016/S0022-1694(99)00058-X

Kumar, R., Musuuza, J.L., Van Loon, A.F., Teuling, A.J., Barthel, R., Ten Broek, J., Mai, J., Samaniego, L., Attinger, S., 2016. Multiscale evaluation of the Standardized Precipitation Index as a groundwater drought indicator. Hydrol. Earth Syst. Sci. 20, 1117–1131. doi:10.5194/hess-20-1117-2016

Long, A.J., Mahler, B.J., 2013. Prediction, time variance, and classification of hydraulic response to recharge in two karst aquifers. Hydrol. Earth Syst. Sci. 17, 281–294. doi:10.5194/hess-17-281-2013

Pinault, J.-L., Pauwels, H., Cann, C., 2001. Inverse modeling of the hydrological and the hydrochemical behavior of hydrosystems: Application to nitrate transport and denitrification. Water Resour. Res. 37, 2179–2190.

---

## Referee Comment (RC1) · A.J. Long (Referee) · 18 Jan 2017

This manuscript describes the application of an existing hydrologic model for karst aquifers. The approach of evaluating the model calibration in terms of hydrologic exceedance rates appears to be a new and useful approach. Exceedance rates of projected hydrologic simulations are evaluated for human safety or the needs of species (e.g., https://pubs.er.usgs.gov/publication/sir20145089). Therefore, if a model is to be used for this purpose, it makes good sense to evaluate the model directly on the basis of exceedance frequencies.

Main comments:

[Figure]

The abstract explains that the approach to simulate groundwater level frequency is novel. I would say that this is not the novel part, because the time-series records were simulated and simply converted into frequency distributions, which is a common way to summarize hydrologic time-series records. However, the novel part is that the model calibration is evaluated on the basis of frequency distributions, which I have not seen before, and I suggest presenting it that way. The tile is more accurate: "A percentile approach to evaluate simulated groundwater levels and frequencies..."

Section 5.2 discusses the possibility of focusing the calibration on high percentiles. I don't totally agree that longer time-series records would be needed to do this, and this section could benefit from further discussion of this idea. For example, an approach could be developed to evaluate the usefulness and data adequacy of such an endeavor. You could vary the weights within the observed time-series record for individual observations at different exceedances to tailor the calibration to a target percentile. It would be possible to calibrate to a different weighting scheme for each percentile. Further, an uncertainty analysis could be applied on each separate calibration run, and quantifying the presumed decrease in uncertainty as the percentile increases could be useful. Then, when you make predictions for different percentiles, you can also report the differences in uncertainty. This idea also applies to section 5.3, which discusses the prediction of increased drought.

The Introduction discusses risks to events such as groundwater flooding and drought. I suggest adding a short statement to this effect in the Abstract to emphasize the need for this study in terms of natural hazards.

Other comments:

1. p. 2, lines 24-26: indicates that karst groundwater levels were simulated by lumped models only in a few instances, but see also Long and Derickson (1999), Long and Mahler (2013), and Pinault et al. (2001). 2. p. 3, lines 22-23: describes a new approach to show groundwater levels as frequency distributions. Showing hydrologic

time-series data as frequency distributions is a common method. Please explain how this is new, or describe it differently. 3. p. 3, line 35: "PET" should be defined. 4. p. 4, line 34: discusses a "weighting scheme." I think the calibration weights are applied to observations, but that should be explained here for clarity. 5. figure 7: what is the meaning of "manipulated" in the caption? 6. table 5: I think these result apply to a particular model time step (e.g., daily), but I'm not sure. Please clarify.

References:

Long, A.J., Derickson, R.G., 1999. Linear systems analysis in a karst aquifer. J. Hydrol. 219, 206-217.

Long, A.J., Mahler, B.J., 2013. Prediction, time variance, and classification of hydraulic response to recharge in two karst aquifers. Hydrol. Earth Syst. Sci. 17, 281-94.

Pinault, J.L., Plagnes, V., Aquilina, L., Bakalowicz, M., 2001. Inverse modeling of the hydrological and the hydrochemical behavior of hydrosystems; characterization of karst system functioning. Water Resour.Res. 37, 2191-2204.

---

## Referee Comment (RC2) · Anonymous Referee #2 · 16 Feb 2017

A percentile approach to evaluate simulated groundwater levels and frequencies in a Chalk catchment in Southwest England.

The authors use the VarKarst model to predict the variation of discharge and groundwater levels in a catchment in England. The topic is relevant to the journal and the work is timely given a growing interest in the forecasting and characterisation of floods and droughts. It would be very valuable to have a discharge/groundwater level model that gives reliable predictions even when the calibration datasets are small. The paper is suitably concise and the description is generally clear. However, I have a number of serious concerns about the focus of the manuscript and the calculations within it. I am unable to recommend the manuscript for publication unless these concerns are

addressed.

In the title and introduction, the authors promote their 'percentile approach' to assessing the performance of the models as the main novelty in the manuscript. I am afraid that I am not persuaded that the percentile approach is novel enough to merit publication in itself. The approach is a comparison between the realised percentiles of the observed and modelled discharge/groundwater levels. It appears to be exactly equivalent to the standard statistical procedure of comparing the distributions of two variables in terms of their realized quantiles. This is a very well used approach, as evidenced by the Wikipedia page describing the QQ plots that result:

https://en.wikipedia.org/wiki/Q%E2%80%93Q_plot

Furthermore, I am not convinced that the percentiles used by the authors are a good indicator of the performance of a discharge/groundwater level model. The authors are only confirming that the complete set of modelled values are similar to the complete set of observed values. They are not confirming that the groundwater levels are predicted at the correct time. In terms of the authors' percentile criterion, there would be no penalty for a model that predicts a flood at the time of a drought but compensates by predicting a drought at the time of a flood. For these reasons, I believe that a substantial change of theme of the manuscript is required.

The theme that most interests me in the manuscript is the quest to "balance model complexity and data availability" referred to in the Abstract. If the authors could demonstrate that they have achieved this for their study area then they would have a very valuable paper. However, I believe that much more evidence of this is required.

The authors calibrate the 13 parameters of the VarKarst model using data from three boreholes and one timeseries of discharge data. In any such modelling exercise I am concerned whether the parameters maintain their physical meaning and whether the internal processes in the model (e.g. the soil and epikarst modules) are reflecting reality. It is entirely possible that the model is acting as a 'black box' where the large
number of parameters are giving it the flexibility to reproduce almost any relationship between the input and output data with which it is presented. If this were the case, it is unlikely that the model would perform well if the characteristics of the input data were to change (e.g. under climate change).

One piece of evidence of the model reflecting reality rather than acting as a black box would be clearly identifiable parameter values. The authors are therefore quite correct to explore the identifabilty of the parameters using the MCMC approach. Their results (Figure 5) indicate that for their final calibration that the parameters are almost perfectly identifiable. Given the short duration, high seasonality and marked temporal correlation amongst the input data I find this surprising. Indeed when (Schoups and Vrugt, 2010) calibrated their similarly complex river models using an MCMC approach many of the parameter values could not be identified. This makes me question the authors' implementation of the MCMC approach.

Within a MCMC algorithm, a huge number of different sets of parameter values are compared. Those sets that are consistent with the observed data are included in the Markov chain whereas other parameter sets are discarded. These comparisons are normally made by calculating the likelihood function for the different parameter sets (e.g. Schoups & Vrugt, 2010). It is possible to use the calculated likelihoods or probabilities to determine which parameters sets are good enough to be included in the Markov chain. Thus, the inclusion or exclusion of a parameter set is decided by an objective criterion that is consistent with statistical theory.

It appears that the authors have compared different parameter sets in terms of their KGE score. This concerns me because it is not clear to me how to decide what magnitude of difference between KGE scores signifies that one set of parameters is not good enough to be included. A threshold on the KGE scores could be set arbitrarily but then the realised distributions of the parameters become meaningless. The apparent identifiability of the parameters could be changed by a simple and arbitrary tweak of this threshold.
[Figure]

Therefore, the authors must give more detail about the comparison function they included in the MCMC algorithm and demonstrate how it leads to objective estimates of the posterior distributions of the parameters.

I'd also like clarification about how the authors decided that their validation results were sufficiently good to conclude that "the model provides robust simulations of discharge and groundwater levels". The authors state that the difference between the calibration and validation KGE scores are small. For each data source, the validation results are worse than the calibration results. Might this indicate that the model is too complex? How big a difference between validation and calibration results would have been required for the authors to conclude that the model had been ineffective? There is a great deal of seasonality in the groundwater levels. Can we be sure that the model is going beyond these seasonal trends? Could a simple annual periodic function have given similarly good results and better managed the trade-off between model complexity and data availability?

Specific comments:

The introduction provides a clear description of the hydrogeological system with the appropriate level of description and ample references for anyone who wants to delve further (the same can also be said of section 2). More detail could be provided in the paragraph which describes the importance of the work in this study.

I appreciate that the authors have made the Methodology section concise by referring to previous papers. However, I think they could give a clearer overview of the VarKarst model whilst leaving the details to the other papers. What do the 15 model compartments correspond to? Are they situated along some sort of gradient in the catchment? If so, is it possible to use knowledge of the hydrogeological system to determine the compartment in which each borehole is situated? What do they mean when they say that the spatial variability of the soil, epikarst and groundwater systems are expressed as a Pareto function? - What characteristics of these systems are the authors referring

to? - Are these characteristics sampled from a Pareto distribution or do they decay according to a Pareto function?

Equation (1). Ensure that all symbols in all equations are defined. Use a multiplication sign rather than '*'.

Section 3.3 – Give more detail about the implementation of the MCMC algorithm to address my concerns above. In particular, explicitly state the function used to decide whether a parameter set is accepted or rejected and explain how these lead to objective and representative samples of the posterior distributions.

Section 3.4 The authors state that their percentile approach was motivated by standardised groundwater and precipitation indices. Seasonality is often removed from standardised indices. Did the authors consider removing seasonality from their simulations before assessing them?

Equation (2) Write words such as mean in standard font rather than italics. State which variable you are summing over.

The authors calculated the 5th percentile at a yearly time scale using 10 years of data. Does this mean they attempted to determine the 5th percentile from only 10 observations of yearly data?

Section 3.5: I am not sure that using nine climate scenarios is sufficient to assess the uncertainty in the effect of climate change on groundwater levels.

Section 4.2. The poor performance of the model when groundwater levels are large could be because the authors are using an objective function that is suited to Normally distributed variables but the distribution of groundwater levels are skewed. Have the authors tried an objective function that is more suited to skewed data?

References

Schoups, G., Vrugt, J.A., 2010. A formal likelihood function for parameter and predictive inference of hydrologic models with correlated, heteroscedastic, and non-Gaussian errors. Water Resources Research, 46, W10531.

---

## Short Comment (SC2) · 19 Mar 2017

We would like to refer to the attached response file.

Please also note the supplement to this comment:
http://www.nat-hazards-earth-syst-sci-discuss.net/nhess-2016-386/nhess-2016-386-SC2-supplement.pdf

---

## Author Response (AR1)

**Response Letter**

Dear editor,

Dear reviewers,

Many thanks for your valuable comments and suggestions. In the revised manuscript, we incorporated the suggested changes according to our response in the open discussion. Please find below a point by point response on all your comments followed by the revised manuscript with tracked changes.

We hope that the revised manuscript will be satisfactory to be published in Natural Hazards and Earth System Sciences.

Sincerely,

Simon Brenner on behalf of the co-authors.

**Associate Editor**

*Based upon the reviews, the article needs major revision. Authors are kindly invited to follow the indications by the reviewers when preparing the revised manuscript, or, in case they disagree with their comments, to explain in detail the reasons why.*

We thank the associate editor and the two reviewers for their detailed evaluation.

**Referee Andrew Long**

*This manuscript describes the application of an existing hydrologic model for karst aquifers. The approach of evaluating the model calibration in terms of hydrologic exceedance rates appears to be a new and useful approach. Exceedance rates of projected hydrologic simulations are evaluated for human safety or the needs of species (e.g., https://pubs.er.usgs.gov/publication/sir20145089). Therefore, if a model is to be used for this purpose, it makes good sense to evaluate the model directly on the basis of exceedance frequencies.*

We thank Dr Andy Long for his positive evaluation and his valuable and helpful comments.

*Main comments:*

*1. The abstract explains that the approach to simulate groundwater level frequency is novel. I would say that this is not the novel part, because the time-series records were simulated and simply converted into frequency distributions, which is a common way to summarize hydrologic time-series records. However, the novel part is that the model calibration is evaluated on the basis of frequency distributions, which I have not seen before, and I suggest presenting it that way. The tile is more accurate: "A percentile approach to evaluate simulated groundwater levels and frequencies. . ."*

We changed the title to "Process-based modelling to evaluate simulated groundwater levels and frequencies in a Chalk catchment in Southwest England" and improved the abstract accordingly.

*2. Section 5.2 discusses the possibility of focusing the calibration on high percentiles. I don't totally agree that longer time-series records would be needed to do this, and this section could benefit from further discussion of this idea. For example, an approach could be developed to evaluate the usefulness and data adequacy of such an endeavor. You could vary the weights within the observed time-series record for individual observations at different exceedances to tailor the calibration to a target percentile. It would be possible to calibrate to a different weighting scheme for each percentile. Further, an uncertainty analysis could be applied on each separate calibration run, and quantifying the presumed decrease in uncertainty as the percentile increases could be useful. Then, when you make predictions for different percentiles, you can also report the differences in uncertainty. This idea also applies to section 5.3, which discusses the prediction of increased drought.*

We added a statement at the end of Section 5.2:

"This could be further evaluated by using different percentile weighting schemes, stepwise increasing the weight on the target percentile."

*3. The Introduction discusses risks to events such as groundwater flooding and drought. I suggest adding a short statement to this effect in the Abstract to emphasize the need for this study in terms of natural hazards.*

We added the sentence: "Due to their properties, they are particularly vulnerable to groundwater related hazards like floods and droughts."

*Other comments:*

*1. p. 2, lines 24-26: indicates that karst groundwater levels were simulated by lumped models only in a few instances, but see also Long and Derickson (1999), Long and Mahler (2013), and Pinault et al. (2001).*

We added Long and Mahler (2013) and (Derickson and Long (1999)) to the literature comparison. We could not find any simulation of groundwater levels in the work of Pinault et al. (2001).

*2. p. 3, lines 22-23: describes a new approach to show groundwater levels as frequency distributions. Showing hydrologic time-series data as frequency distributions is a common method. Please explain how this is new, or describe it differently.*

*3. p. 3, line 35: "PET" should be defined.*

We agree with both comments and improved the text accordingly.

*4. p. 4, line 34: discusses a "weighting scheme." I think the calibration weights are applied to observations, but that should be explained here for clarity.*

We added "(…), as we stepwise added borehole data to our discharge observations." and further added more detailed information about the model calibration in section 3.3 emphasizing the importance of auxiliary data.

*5. figure 7: what is the meaning of "manipulated" in the caption?*

This refers to the manipulation of our observed "baseline" data, see chapter 3.5. We clarified the caption of figure 7.

*6. table 5: I think these result apply to a particular model time step (e.g., daily), but I'm not sure. Please clarify.*

These are the mean model outputs (Qsim, AET) and exceedances per year in the simulation period 2070-2099 which was calculated on the basis of our model which runs at a daily time step. We added:

" They display the mean model outputs (Qsim, AET) and mean exceedances per year, calculated on the basis of our modelled time series. "

in section 4.3

**References**

Derickson, R.., Long,   a. ., 1999. Linear systems analysis in a karst aquifer. J. Hydrol. 219, 206–217. doi:10.1016/S0022-1694(99)00058-X

Kumar, R., Musuuza, J.L., Van Loon, A.F., Teuling, A.J., Barthel, R., Ten Broek, J., Mai, J., Samaniego, L., Attinger, S., 2016. Multiscale evaluation of the Standardized Precipitation Index as a groundwater drought indicator. Hydrol. Earth Syst. Sci. 20, 1117–1131. doi:10.5194/hess-20-1117-2016

Long, A.J., Mahler, B.J., 2013. Prediction, time variance, and classification of hydraulic response to recharge in two karst aquifers. Hydrol. Earth Syst. Sci. 17, 281–294. doi:10.5194/hess-17-281-2013

Pinault, J.-L., Pauwels, H., Cann, C., 2001. Inverse modeling of the hydrological and the hydrochemical behavior of hydrosystems: Application to nitrate transport and denitrification. Water Resour. Res. 37, 2179–2190.

**Anonymous Referee**

*A percentile approach to evaluate simulated groundwater levels and frequencies in a Chalk catchment in Southwest England. The authors use the VarKarst model to predict the variation of discharge and groundwater levels in a catchment in England. The topic is relevant to the journal and the work is timely given a growing interest in the forecasting and characterisation of floods and droughts. It would be very valuable to have a discharge/groundwater level model that gives reliable predictions even when the calibration datasets are small. The paper is suitably concise and the description is generally clear. However, I have a number of serious concerns about the focus of the manuscript and the calculations within it. I am unable to recommend the manuscript for publication unless these concerns are addressed.*

We thank the referee for her /his evaluation and detailed comments. We appreciate the concerns about our claims about the novelty of the approach. We hope that we can clarify all of the referees concerns in the following response.

1. *In the title and introduction, the authors promote their 'percentile approach' to assessing the performance of the models as the main novelty in the manuscript. I am afraid that I am not persuaded that the percentile approach is novel enough to merit publication in itself. The approach is a comparison between the realised percentiles of the observed and modelled discharge/groundwater levels. It appears to be exactly equivalent to the standard statistical procedure of comparing the distributions of two variables in terms of their realized quantiles. This is a very well used approach, as evidenced by the Wikipedia page describing the QQ plots that result: https://en.wikipedia.org/wiki/Q%E2%80%93Q_plot*

The referee is right. The novelty of our research is the application of a process-based model instead of a statistical distribution function. We admittedly created a wrong perception by the choice of our title. The revised manuscript will be titled with "Process-based modelling to evaluate simulated groundwater levels and frequencies in a Chalk catchment in Southwest England "
In addition, we provided more reference to the work of others that applied quantile-quantile approaches in groundwater frequency analysis in the introduction.

2. *Furthermore, I am not convinced that the percentiles used by the authors are a good indicator of the performance of a discharge/groundwater level model. The authors are only confirming that the complete set of modelled values are similar to the complete set of observed values. They are not confirming that the groundwater levels are predicted at the correct time. In terms of the authors' percentile criterion, there would be no penalty for a model that predicts a flood at the time of a drought but compensates by predicting a drought at the time of a flood. For these reasons, I believe that a substantial change of theme of the manuscript is required.*

We believe this is a misunderstanding: In this study, we used the percentile approach only for our evaluation. The calibration and evaluation of our model was carried out with continuous flow and water level observations. The error function KGE that we used for comparing model simulations and observations explicitly evaluates the correctness of timing by using the linear correlation coefficient *r* as one of its three components (see also our response to general comment 5).
In the new version of the manuscript we clarified the elaboration of the model calibration and evaluation (see Section 3.3) to avoid further misunderstanding.

3. *The theme that most interests me in the manuscript is the quest to "balance model complexity and data availability" referred to in the Abstract. If the authors could demonstrate that they have achieved this for their study area then they would have a very valuable paper. However, I believe that much more evidence of this is required. The authors calibrate the 13 parameters of the VarKarst model using data from three boreholes and one timeseries of discharge data. In any such modelling exercise I am concerned whether the parameters maintain their physical meaning and whether the internal processes in the model (e.g. the soil and epikarst modules) are reflecting reality. It is entirely possible that the model is acting as a 'black box' where the large number of parameters are giving it the flexibility to reproduce almost any relationship between the input and output*

*data with which it is presented. If this were the case, it is unlikely that the model would perform well if the characteristics of the input data were to change (e.g. under climate change).*

We understand the concern of the referee. Indeed, models with more than 5-6 parameters are often regarded to end up in equifinality (Jakeman and Hornberger, 1993; Wheater et al., 1986; Ye et al., 1997), i.e. their parameters lose their identifiability (Beven, 2006; Wagener et al., 2002). In such case, the model can be regarded as a "black box" with rather limited prediction skills as correctly stated by the referee.

In order to reflect the complexity of karst hydrology, 5-6 parameters are often not enough to include all relevant processes in a simulation model. For that reason, recent research took advantage of auxiliary data, such as water quality data or tracer experiments (Hartmann et al., 2013; Oehlmann et al., 2015). These studies could show that such information allowed for identifying the necessary model parameters therefore enabling the model to reflect the relevant processes.

In this study, we followed this idea and used a combination of groundwater level observations at three locations and discharge observations to obtain enough information to estimate our model parameters. Applying the Shuffled Complex Evolution Metropolis algorithm (also see our response to general comment 5 below) and step-wise increasing the calibration data (only discharge, only groundwater, all together), we show that discharge alone, as well as groundwater alone, do not provide enough information to identify all of our model parameters (Fig 5 in the manuscript) as the posteriors of some of the model parameters remain close to a uniform distribution.

Using all information, observed discharge and observations of three groundwater levels, all model parameters are identifiable, i.e. their posteriors strongly differ from a uniform distribution (blue lines in Fig 5), which is in accordance with preceding research that showed that a combination of groundwater and discharge observations can parameter uncertainty (Kuczera and Mroczkowski, 1998). Furthermore, the split-sample test indicates a stable performance of groundwater simulations (Table 3, also see our response to general comment 6). We therefore believe that there is enough indication that the model reproduces the system behaviour satisfactorily and that it can be used for prediction.

We added these clarifications to the methodology section (3.3) as well as in the discussion (5.1)..

4. *One piece of evidence of the model reflecting reality rather than acting as a black box would be clearly identifiable parameter values. The authors are therefore quite correct to explore the identifabilty of the parameters using the MCMC approach. Their results (Figure 5) indicate that for their final calibration that the parameters are almost perfectly identifiable. Given the short duration, high seasonality and marked temporal correlation amongst the input data I find this surprising. Indeed when (Schoups and Vrugt, 2010) calibrated their similarly complex river models using an MCMC approach many of the parameter values could not be identified. This makes me question the authors' implementation of the MCMC approach.*

Thanks for this critical comment. We thoroughly studied the work of Schoups and Vrugt (2010) in relation to our results. Using a hydrological model with seven parameters combined with an error model with 4-5 parameters their calibration problem is indeed similar to the one we present in our study. But there is one important difference: they only use discharge observations for model calibration. As found by many preceding studies (see also our response to general comment 2) simulation models with more than 5-6 parameters typically result in increased parameter uncertainty, which Schoups and Vrugt (2010) also found in their study. Using only discharge information, our study would have resulted in similar problems (green lines in Fig 5). However, the combined use of discharge observations and the observations of three groundwater wells resulted in increased parameter identifiability, as we could also show in Fig 5 (blue lines). Therefore, our results do not contradict Schoups and Vrugt (2010) but they rather show that there are ways to reduce parameter uncertainty by auxiliary data (see also our response to general comment 2).

In the revised manuscript, we put more emphasis on the description of these multiple data sets for parameters estimation (Section 3.3) and refer to the comparison with Schoups and Vrugt (2010) in the discussion (Section 5.1).

5.  *Within a MCMC algorithm, a huge number of different sets of parameter values are compared. Those sets that are consistent with the observed data are included in the Markov chain whereas other parameter sets are discarded. These comparisons are normally made by calculating the likelihood function for the different parameter sets (e.g. Schoups & Vrugt, 2010). It is possible to use the calculated likelihoods or probabilities to determine which parameters sets are good enough to be included in the Markov chain. Thus, the inclusion or exclusion of a parameter set is decided by an objective criterion that is consistent with statistical theory.*

    *It appears that the authors have compared different parameter sets in terms of their KGE score. This concerns me because it is not clear to me how to decide what magnitude of difference between KGE scores signifies that one set of parameters is not good enough to be included. A threshold on the KGE scores could be set arbitrarily but then the realised distributions of the parameters become meaningless. The apparent identifiability of the parameters could be changed by a simple and arbitrary tweak of this threshold.*

    *Therefore, the authors must give more detail about the comparison function they included in the MCMC algorithm and demonstrate how it leads to objective estimates of the posterior distributions of the parameters.*

The Shuffled Complex Evolution Metropolis algorithm (SCEM, Vrugt et al., 2003) that we used in our study is based on the Metropolis-Hastings algorithm (Hastings, 1970; Metropolis et al., 1953) and the Shuffled Complex Evolution algorithm (Duan et al., 1992). The Metropolis-Hastings algorithm uses a formal likelihood measure, i.e. an objective criterion that is consistent with statistical theory, and calculates the ratio of the posterior probability densities of a "candidate" parameter set that is drawn from a proposal distribution and a given parameter set. If this ratio is larger or equal than a number randomly drawn from a uniform distribution between 0 and 1, the "candidate" parameter set is accepted. This procedure is repeated for a large number of iterations. If the proposal distribution is properly chosen, the Markov Chain will rapidly explore the parameter space and it will converge to the target distribution of interest (Vrugt et al., 2003).

In the SCEM algorithm, "candidate" parameter sets are drawn from a self-adapting proposal distribution for each of a predefined number of clusters. Again a random number [0,1] is used to accept or discard "candidate" parameter sets. In our study, we use the Kling-Gupta efficiency KGE (Gupta et al., 2009) as the objective function, which can be regarded as an informal likelihood measure (Smith et al., 2008). To decide whether to accept or discard a parameter set, we compare the KGEs of the "candidate" and the given parameter sets. Such procedure was already applied in various studies (Blasone et al., 2008; Engeland et al., 2005; McMillan and Clark, 2009) and is possible if the error functions monotonically increasing with improved performance. We achieved this in the SCEM algorithm by defining $KGE_{SCEM}$ as

$$KGE_{SCEM} = -\sqrt{(r-1)^2 + (\alpha-1)^2 + (\beta-1)^2}$$
$$\alpha = \frac{\sigma_S}{\sigma_O}; \beta = \frac{\mu_S}{\mu_O}$$

With $r$ as the linear correlation coefficient between simulations and observations, and $\sigma_S$, $\sigma_O$ and $\mu_S$, $\mu_O$ as the means and standard deviations of simulations and observations, respectively.

As stated correctly by the referee, the shape of the posteriors is dependent on the error function and using another likelihood measure, formal or informal, may have resulted in different shapes of the posteriors. However, applying SCEM with KGE in our stepwise procedure we are mostly interested in the relative differences of the posteriors and we can clearly see how some of posteriors translate from a uniform distribution to a well-defined peak when more information is added (see also our response to general comment 4). These results combined with the acceptable multi-objective performance of the model during calibration and validation (see also our response to general comment 6), and the realistic parameters that we finally found (see discussion of parameter values in subsection 5.3) makes us confident that the model reproduces the relevant features of our studied system.

We added these elaborations in the methods section and discuss the consequences of using an informal likelihood measure in the revised discussion.

6. *I'd also like clarification about how the authors decided that their validation results were sufficiently good to conclude that "the model provides robust simulations of discharge and groundwater levels". The authors state that the difference between the calibration and validation KGE scores are small. For each data source, the validation results are worse than the calibration results. Might this indicate that the model is too complex? How big a difference between validation and calibration results would have been required for the authors to conclude that the model had been ineffective?*

Split-sample tests are a common and necessary tool to evaluate the prediction performance of a simulation model (Klemeš, 1986). If the model is compared to a validation period, i.e. a time series of observations that was not used for parameters estimation, a decrease of performance has to be expected because there is always a tendency to compensate for model structural limitations and observational uncertainties during the calibration. If a model contains too many degrees of freedom (model parameters), there is a risk that calibration may overcome all these limitations and uncertainties although the model is a poor choice for the studied system. A split sample-test would indicate such failure by a strong decrease of performance during the validation period.

As correctly questioned by the referee the threshold, from which a decrease of performance is not acceptable anymore, is subject to the individual case of application and the opinion of the modeller. In our case we obtained a decrease of performance from -11% (groundwater prediction) to -21% (discharge prediction). Such ranges are comparable with split-sample tests found by other studies (-4% to -14% by Parajka et al., 2007; -5% to -24% by Perrin et al., 2001). The lower decrease in performance that we found for the simulation of groundwater levels also indicates more stable prediction performance for the groundwater simulations that we later use for our example application with the simplified climate scenarios.

We added these aspects to the discussion (Section 5.1).

7. *There is a great deal of seasonality in the groundwater levels. Can we be sure that the model is going beyond these seasonal trends? Could a simple annual periodic function have given similarly good results and better managed the trade-off between model complexity and data availability?*

Yes, a simple annual periodic function may be able to reproduce observed variability of groundwater levels to some degree. However, such function would not be more than a black box model and it would not be straight forward using it to assess the impact of climatic changes on groundwater levels. The structure of the VarKarst model takes into account the particulates of karst hydrology (see also our response to specific comment 1). We believe that our analysis and evaluation provides some indication that it is also able to reflect the observed processes at our Chalk study site, therefore making it a useful tool to explore the impact of climate changes on groundwater level dynamics.

We added:

*" However, present approaches mostly rely on statistical distribution functions to express groundwater dynamics and groundwater level exceedance probabilities (e.g., Bloomfield et al., 2015; Kumar et al., 2016) and it is questionable whether the shapes of these distribution functions remain the same when climate or land use change."*

to the introduction section.

Specific comments:

1. *The introduction provides a clear description of the hydrogeological system with the appropriate level of description and ample references for anyone who wants to delve further (the same can also be said of section 2). More detail could be provided in the paragraph which describes the importance of the work in this study. I appreciate that the authors have made the Methodology section concise by referring to previous papers. However, I think they could give a clearer overview of the VarKarst model whilst leaving the details to the other papers. What do the 15 model compartments correspond to? Are they situated along some sort of gradient in the catchment? If so, is it possible to use knowledge of the hydrogeological system to*

*determine the compartment in which each borehole is situated? What do they mean when they say that the spatial variability of the soil, epikarst and groundwater systems are expressed as a Pareto function? - What characteristics of these systems are the authors referring to? - Are these characteristics sampled from a Pareto distribution or do they decay according to a Pareto function?*

We thank the referee for these helpful suggestions. We added statements to the abstract and the introduction to highlight the scope and importance of this work (see also general comment 7 above as well as the comment 3 of the other referee).

In addition, we added a more detailed description of the VarKarst model and the meaning of its individual components (see Appendix).

2. *Equation (1). Ensure that all symbols in all equations are defined. Use a multiplication sign rather than '*'.*

We improved our manuscript and eliminated the stylistic flaws.

3. *Section 3.3 – Give more detail about the implementation of the MCMC algorithm to address my concerns above. In particular, explicitly state the function used to decide whether a parameter set is accepted or rejected and explain how these lead to objective and representative samples of the posterior distributions.*

Please see our response to general comment 4.

4. *Section 3.4 The authors state that their percentile approach was motivated by standardised groundwater and precipitation indices. Seasonality is often removed from standardised indices. Did the authors consider removing seasonality from their simulations before assessing them?*

During the period of model development and calibration, we considered calibrating the flow percentiles directly, i.e. removing seasonality. However, removing the temporal information from the time series would have reduced the information content of the data and would have resulted in increased parameter uncertainty (see our response to general comment 2 and 3) with a lower prediction performance of the model.
We added this information at the end of the section 3.4 .

5. *Equation (2) Write words such as mean in standard font rather than italics. State which variable you are summing over. The authors calculated the 5th percentile at a yearly time scale using 10 years of data. Does this mean they attempted to determine the 5th percentile from only 10 observations of yearly data?*

We improved the elaborations on Eq. 2.
The percentiles were derived from the daily data of the calibration period (2008-2012). We then compared the average sum of days exceeding the respective percentile in the respective time scale. We added a sentence to the end of section 3.4 to clarify the origin of the percentiles.

6. *Section 3.5: I am not sure that using nine climate scenarios is sufficient to assess the uncertainty in the effect of climate change on groundwater levels.*

The purpose of the simple climate scenarios was to provide an application example of the new methodology, which is rather hypothetical considering the large uncertainties of current climate projections. We believe that our 9 realisations are sufficient to show that different possible future changes have a non-linear impact on groundwater level frequencies.
We added this elaboration to section 5.3

*7. Section 4.2. The poor performance of the model when groundwater levels are large could be because the authors are using an objective function that is suited to Normally distributed variables but the distribution of groundwater levels are skewed. Have the authors tried an objective function that is more suited to skewed data?*

The objective function (KGE) is applied to simulated time series of groundwater levels. 
[revised manuscript text omitted]

---

## Author Response (AR2)

**Response Letter**

Dear editor,

Dear reviewers,

Thank you very much for your valuable comments and suggestions. In the re-revised manuscript, we incorporated all the comments of referee #3. We respectfully disagree with the negative evaluation of referee #2, which mostly repeated the criticism from her/his last review. Nevertheless, some clarification was added to the manuscript following her/his criticism. Please see our point-by-point response below for more details.

We hope that the revised manuscript will be satisfactory to be published in Natural Hazards and Earth System Sciences.

Sincerely,

Simon Brenner on behalf of the co-authors.

**Associate Editor**

*Minor revisions are required to your article. Please, when preparing the revised edition, take into the due account all comments and suggestions from the reviewers. In case you disagree with some of them, please indicate the reasons why. I am looking forward to receiving your revised paper.*

We thank the associate editor for his positive evaluation.

**Referee #3**

*The manuscript from Brenner et al. is dealing with modeling of fractured and karstified aquifers in England. I found the topic well developed and of interest for an international audience. The contribution is well organized and contains interesting data and discussion. Some minor comments are detailed below:*

We thank the referee #3 for his positive comment on our manuscript.

*- Introduction: the literature review on this topic is wider than described in the text. About potential future changes in groundwater dynamics (lines 14-15 p2), several additional examples can be found in recent literature; please add references using recent papers*

We agree and added the following papers dealing with the investigation of groundwater dynamics (P2L13-16 in the revised manuscript):

Perrone, D. and Jasechko, S.: Dry groundwater wells in the western United States, Environ. Res. Lett., 12(10), 104002 [online] Available from: http://stacks.iop.org/1748-9326/12/i=10/a=104002, 2017.

Naughton, O., Johnston, P. M., Mccormack, T. and Gill, L. W.: Groundwater flood risk mapping and management : examples from a lowland karst catchment in Ireland, , doi:10.1111/jfr3.12145, 2015.

Moutahir, H., Bellot, P., Monjo, R., Bellot, J., Garcia, M. and Touhami, I.: Likely effects of climate change on groundwater availability in a Mediterranean region of Southeastern Spain, , 176(October 2016), 161–176, doi:10.1002/hyp.10988, 2017.

von Freyberg, J., Moeck, C. and Schirmer, M.: Estimation of groundwater recharge and drought severity with varying model complexity, J. Hydrol., 527(Supplement C), 844–857, doi:https://doi.org/10.1016/j.jhydrol.2015.05.025, 2015.

*- Results: what are the "hardly identifiable parameters"? See line 22 p7. Please list the parameters which are not well identifiable and discuss this limit in the discussion section*

*- Discussion: same topic, line 38 p8: "all model parameters are identifiable": It does not seem from the figure*

The parameters are unidentifiable when we use only discharge or the groundwater time series. When using all information, the cumulative parameter distributions show identifiability throughout all parameters (blue lines). We added some clarification in both sections (P7L27-34 in the revised manuscript).

*- line 26-27 p9: the sentence starting with "This is obvious" is not clear to me, please revise and explain better.*

We apologize for the confusing sentence. We improved it.

*- Conclusions: this chapter needs revision; at the moment it appears more like an abstract than a conclusion. Please identify the main findings and the main limits of your study, and list if possible as "take home messages", clearly and concisely.*

We agree and added some sentences about highlighting the key aspects and limits of our findings (P11L13-16 in the revised manuscript).

**Referee #2**

*This is a revised version of a manuscript detailing the simulation of groundwater levels in a Karst environment in the Southwest of England. I described in my review of the original manuscript how this topic is highly relevant to the journal and timely given the increasing need to simulate and forecast groundwater levels from limited datasets. The manuscript is suitably concise and the description is clear. I fully agree with the authors' statement in their Abstract that "specialised modelling approaches*

*are required that balance model complexity and data availability". The authors assess whether they have achieved this balance by both exploring the identifiability of the parameters within their model (using a Shuffled Complex Evolution Method; SCEM) and by comparing model performance metrics for calibration and validation datasets (i.e. a split-sample test). They conclude that their modelling exercise had been a success because their analyses suggest that all of the parameters are identifiable and the differences between the calibration and validation metrics take values which they consider to be small.*

*Whilst I fully endorse the authors' general approach to assessing the performance of their model I have severe concerns about the exact way in which it has been implemented. I do not believe that the posterior distributions of the parameters yielded from the SCEM accurately reflect the uncertainty of these parameters. Furthermore, I do not believe that the comparison between*

*performance metrics upon calibration and validation are particularly meaningful. For these reasons I do not recommend that the manuscript is accepted for publication.*

We respectfully disagree with the opinion of referee #2. We already provided a thorough discussion and reply on her/his criticism concerning the reliability of the parameter posteriors, the choice of objective function and the model validation in the
first round of review trying to incorporate the constructive elements of the review. For that reason, we will keep our response to this second review a bit shorter.

*I first detail my concerns about the analyses of parameter identifiability. Looking at Figure 5, it is apparent that according to the SCEM that when the model is calibrated using only discharge data that the Kc parameter (for example) is almost perfectly*
*identifiable. This posterior distribution indicates that this parameter definitely has a value less than 1. However, when all of the calibration data is used the parameter definitely has a value greater than 9. This is a clear contradiction and at least one of these two posterior distributions must be incorrect. Similar contradictions are evident for all of the other parameters except for those related to the groundwater level in a specific borehole.*

We disagree. As we only consider marginal distributions, parameter interactions (see standard literature, e.g. Saltelli et al.,
2008), which may cause an apparent sensitivity of a model parameter, are not visible. Such behaviour is a common feature of hydrological models and should also be familiar to referee #2. We added some elaboration to the discussion of the re-revised manuscript to make sure no other reader gets the wrong impression of a "clear contradiction" (P9L23-25 in the revised manuscript). Our own research, as well as many other colleagues, has shown how models calibrated or sampled using Monte Carlo simulations using different objectives or criteria 'converge' to difference parts of the parameter space, including scenarios
of non-overlapping posterior distributions (i.e. Freer et al., 1996; Freer et al., 2003; Freer et al., 2004; Seibert and McDonnell, 2002, etc.). This is a long standing discussion in hydrology that we have directly participated in a number of times. It remains an important topic 'in general' for environmental model evaluation and calibration, but that is not the focus of this paper to re-air all of that discussion again and we have ongoing tailored research that is addressing those issues. We feel we have done enough to discuss this one feature of our results in the context of this paper.

*Furthermore, the theoretical justification for the authors' choice to use the Kling-Gupta efficiency (KGE) as the objective function within the SCEM is rather weak. The formal theory of Markov Chain Monte Carlo methods such as the SCEM require that the objective function is a likelihood (i.e. the probability that the data is realised from the proposed model). The authors indicate that the KGE can be treated as an 'informal' likelihood function and refer to a paper by Smith, Bevan and Tawn. This*
*paper does discuss informal likelihood functions and describes sufficient conditions for informal likelihood functions to satisfy the most fundamental axioms of a probability. As far as I can see, the paper does not explicitly mention the KGE. The starting point for satisfying the axioms of probability is that the informal likelihood function can be written as an Lp-norm. It is not immediately clear to me that the KGE can be written as an Lp-norm. Therefore, I am unclear of the relevance of the Smith et*

*al. paper to the authors' study and I am not convinced that the KGE satisfies the fundamental axioms of a probability. I would have thought these axioms were a necessary requirement for a function to be treated as a likelihood.*

We disagree, we should note the paper title for Smith et al. (2008) is called '*Informal likelihood measures in model assessment: Theoretic development and investigation*'. It fundamentally discusses the issues of applying such measures for environmental problems that are not well constrained by idealised experimental designs that are better approximated by more classical statistical theory and so reason why such informal approaches have value for these circumstances. What the author is arguing is that they philosophically disagree with an entire branch of accepted literature on these issues in hydrology, some of which have shown the danger of making strong formal assumptions in the likelihood measure when the error residuals are miss-specified due to epistemic uncertainties (such as Beven et al., 2008). We required from our analysis to allow identifying the information brought by the different types of observation data. As long as our performance metric is fit for purpose, in that it is a useful and valuable monotonically increasing measure of model performance that does not overly weight the very best simulation above all others due to our lack of evidence to fully quantify all the epistemic and aleatory uncertainties within our experimental design (i.e. the over constraining problem). The posteriors clearly indicate that when observations from the different wells are added to the discharge observations during the SCEM analysis, the well parameters (p_Offs and P_GW)

switch from insensitive (close to a uniformly distributed posterior) to a sensitive (strongly differing from a uniform distribution) mode. This provides indication that our requirements are fulfilled, even though with some uncertainty in determining the shape of the distributions due to the choice of an informal likelihood measure, which we already acknowledged in subsection 5.1 as a result of the first review of referee #2. As noted previously we have, and do, work directly on these problems of criteria and objective choice and the impact on final model outputs.

*The authors do not provide any calibration diagnostics which might indicate that the SCEM has converged to a stable posterior distribution.*

The algorithm was applied in default mode as described in Vrugt et al., (2003), which also explains convergence is necessary.

A short clarification was added to the methods section (P4, L38-39).

*The authors do not conduct any validation tests which might indicate that the posterior distributions reflect the uncertainty of the model parameters.*

We disagree. In fact, adding iteratively data to the SCEM analysis can be seen as a test of the reliability of the distributions: If the groundwater well parameters (p_Offs and P_GW) would show sensitivity when only using discharge observations in the SCEM analysis, the validation would be negative. But as they remain insensitive until groundwater observations are added to the analysis, we regard the validation to be positive. Please see our comment above, our response to the previous review of referee #2, and the elaborations in the discussion of the manuscript (subsection 5.1.).

*I similarly have a number of concerns about the authors split sample tests. First, the authors conclude that decreases in model performance upon validation of 11 and 21% are sufficiently small to indicate robust model performance. They refer to other studies where a similar decrease in performance was observed. However, the decision that 21% is 'sufficiently small' is completely subjective. The expected decrease will be a complex function of the number of observations and the degree of*

*seasonality, variability and autocorrelation realised by the data. Therefore, the comparison with other studies is irrelevant. This being said, if I were to compare these values with the results of modelling exercises I have previously undertaken then I would consider 21% to be a relatively large decrease.*

As correctly pointed out by the referee, the acceptance or rejection of a parameters set by prediction is subject to the decision of the individual modeller. In our previous response we provided some studies with comparable changes of model performance in the validation period. By providing the % decrease of model performance we leave the decision of having confidence or doubt about the predictions to the reader. In order to account for this, we replaced the term "acceptable robustness of calibrated parameters" with "certain robustness of calibrated parameter" (P8, L39).

*In their 'Responses to comments' document the authors describe how the KGE objective function "was chosen by trial and*

*error comparing the simulation performances during calibration and validation obtained different objective functions (RMSE and other)". This 'trial and error' approach very much concerns me. The validation data should not be involved with the model calibration in any way – this includes the decisions about how the model is calibrated and what objective function is used. In my opinion, the use of the validation data in this manner invalidates the authors' split-sample tests. If the authors have infinite patience it is almost inevitable that they will eventually find a model calibration set up which yields results which*

*they find pleasing. However, this setup is likely to be particular honed to the particular characteristics of the data they have used and is likely to perform less well as other data become available.*

We disagree. Testing, which performance measure is most adequate to the modelling purpose, is a standard procedure of model development (Freer et al., 1996; Freer et al., 2003; Beven, 2003; Wagener, 2004).

[revised manuscript text omitted]

---

## Author Response (AR3)

**Response Letter**

Dear editor,

Thank you very much for your valuable comments and suggestions. In the re-re-revised manuscript, we incorporated all comments and suggestions (see below).

We hope that the revised manuscript will be satisfactory to be published in Natural Hazards and Earth System Sciences.

Sincerely,

Simon Brenner on behalf of the co-authors.

[revised manuscript text omitted]